

# Does total column ozone change during a solar eclipse?

Germar H. Bernhard[1], George T. Janson[2], Scott Simpson[2], Raúl R. Cordero[3,4], Edgardo I. Sepúlveda Araya[5], Jose Jorquera[3], Juan A. Rayas[6], and Randall N. Lind[1]

[1]Biospherical Instruments, Inc., San Diego, 92110, USA
[2]USDA UV-B Monitoring and Research Program, Colorado State University, Fort Collins, CO 80521, USA
[3]Universidad de Santiago de Chile, Santiago, 9170022, Chile
[4]University of Groningen, Leeuwarden, 8911 CE, Netherlands
[5]Department of Chemical and Environmental Engineering, University of Arizona, Tucson, AZ 85721, USA
[6]Centro de Investigaciones en Óptica A.C., León, 37150, Mexico

*Correspondence to*: Germar H. Bernhard (bernhard@biospherical.com)

**Abstract.** Several publications have reported that total column ozone (TCO) may oscillate with an amplitude of up to 10 Dobson Units during a solar eclipse while other researchers have not seen evidence that an eclipse leads to variations in TCO beyond the typical natural variability. Here, we try to resolve these contradictions by measuring short-term (seconds to minutes) variations in TCO using "global" (Sun and sky) and direct-Sun observations in the ultraviolet (UV) range with filter

radiometers (GUVis-3511 and Microtops). Measurements were performed during three solar eclipses: the "Great American Eclipse" of 2024, which was observed in Mazatlán, Mexico, on 8 April 2024; a partial solar eclipse taking place in the United States on 14 October 2023 and observed at Fort Collins, Colorado (40.57° N, 105.10° W); and a total solar eclipse occurring in Antarctica on 4 December 2021 and observed at Union Glacier (79.76° S, 82.84° W). The upper limit of the amplitude of oscillations in TCO observed at Mazatlán, Fort Collins, and Antarctica were 0.7 %, 0.3 %, and 0.03 %,

respectively. The variability at all sites was within that observed during times not affected by an eclipse. The larger variability at Mazatlán is likely due to cirrus clouds occurring throughout the day of the eclipse and the difficulty of separating changes in the ozone layer from cloud effects. These results support the conclusion that a solar eclipse does not lead to variations in TCO of more than ± 2 Dobson Units and likely much less, drawing into question reports of much larger oscillations. In addition to calculating TCO, we also present changes in the spectral irradiance and aerosol optical depth

during eclipses and compare radiation levels observed during totality. The new results augment our understanding of the effect of a solar eclipse on the Earth's upper atmosphere.

## 1 Introduction

Solar eclipses present a rare and unique opportunity to study the solar corona and changes in the atmosphere prompted by the sudden decrease in the solar flux. Other lesser known, and sometimes curious, phenomena include the dissipation of cumulus

clouds (Trees et al., 2024), changes in the surface tension of water (Fuchs et al., 2019), spiders taking down their webs at the onset of totality (Uetz et al., 2010), and many others. Here we investigate the potential effect of solar eclipses on the ozone




layer. After reviewing observations of changes in total column ozone (TCO) performed during the last century and appraising their potential causes, we present new results based on our own observations during two total solar eclipse in Mexico and Antarctica and one partial eclipse in Colorado. The main objective of the paper is to assess whether the strong

effects of a solar eclipse on the stratospheric ozone layer reported in the past could be real or are spurious results arising from measurement artifacts. We also compare spectral irradiances observed during totality of the 2017, 2021, and 2024 total solar eclipses.

## 1.1 Reported changes in total column ozone during solar eclipses

Variations in TCO during a solar eclipse have been observed as early as 1937 (Kawabata, 1937). Since this time, many
authors have reported short-term and longer-lasting fluctuations in TCO during a solar eclipse (e.g., Kazadzis et al., 2007; Mateos et al., 2014; Mims and Mims, 1993; Zerefos et al., 2000; Zerefos et al., 2001; Zerefos et al., 2007; Antón et al., 2010; Kazantzidis et al., 2007). Additional observations before the 1970s were summarized by Fuchs et al. (2019). Results from these studies often indicate either variations in TCO that are symmetrical relative to the time of totality or sporadic short-term fluctuations in TCO before and after totality. Variations range from unrealistic decreases by more than 100 Dobson
Units (DU) over the course of an eclipse (Jerlov et al., 1954) or equally improbable increases by 70 DU within 15 minutes following totality (Chakrabarty et al., 1997).

Several authors acknowledge that changes in TCO are artifacts of their measurements. Zerefos et al. (2000) report unrealistic *decreases* in TCO measurements during an eclipse and attribute those to diffuse sky radiance entering the field of view of Dobson and Brewer spectrophotometers observing the Sun's direct component. Also Kazadzis et al. (2007) ascribe a
drop in TCO by ~ 75 DU measured during the eclipse of 29 March 2006 with a Brewer to this effect. We find this conclusion unconvincing because the direct-to-diffuse ratio does not change appreciatively during an eclipse (Emde and Mayer, 2007; Bernhard and Petkov, 2019), except for a short (< 4 minutes) period before and after totality. Conversely, Bojkov (1968) finds that ozone *increases* by 25–30 DU during the maximum phase of the eclipse and attributes this increase partly to the wavelength-dependence of the solar limb darkening (LD) effect (Sect. 1.2). By applying a correction for this effect, the
increase is reduced to 14 DU. Antón et al. (2010) find opposite behavior in the evolution of TCO during the solar eclipse of 3 October 2005 derived from a Brewer spectrophotometer and a NILU (Norwegian Institute for Air Research) multi-band instrument: while Brewer TCO values increased by about 15 DU, the NILU measurements decreased about 11 DU. The discrepancy is attributed to measurement artifacts. Mateos et al. (2014) observed TCO during the partial eclipse of 3 November 2013 in Badajoz, Spain, with a handheld Microtops-II sunphotometer from Solar Light (Sect. 3.2) and found a
decrease in TCO by 7 DU during the eclipse maximum. Reasons for this decrease were not provided.

Mims and Mims (1993); Zerefos et al. (2000); Zerefos et al. (2001); and Zerefos et al. (2007) reported short-term fluctuations in TCO during an eclipse, which they attributed to gravity waves. As discussed in Sect. 1.3 in more detail, gravity waves result from the reduction of the solar flux caused by the Moon's shadow during solar eclipses. The sudden





drop in atmospheric energy input may induce perturbations in the ionosphere (Zhang et al., 2017) and stratosphere (Colligan et al., 2020). For example, TCO measurements taken during the total solar eclipse of 11 July 1991 "reveal a sequence of 4 and possibly 5 nearly uniformly spaced fluctuations" after totality (Mims and Mims, 1993). The principle fluctuation, which began 700 s after totality, had a peak-to-peak amplitude of 5 DU (1.7 %). This fluctuation was preceded by one and followed by two fluctuations with reduced amplitude. However, also these observations were likely affected by incomplete LD correction, resulting in TCO changes of 26 DU (about 9 %) over the course of the eclipse. Winkler et al. (2001) concluded that it is "very difficult to quantify these instrumentally based effects and to separate them from naturally produced ozone fluctuations."

Kazantzidis et al. (2007) discuss TCO measurements performed with NILU-UV filter radiometers at several locations in Greece during the total solar eclipse of 29 March 2006. They did not observe any periodic fluctuations in TCO and only report a small increase in TCO of about 5 DU as the visible fraction of the Sun decreases from ~ 60 % to ~ 20 %, followed by a pronounced decrease in measured TCO closer to totality, which they attribute to artifacts of the irradiance measurements.

The clearest evidence to date that a solar eclipse may lead to fluctuation in TCO were presented by Zerefos et al. (2007) who correlated ground-based measurements of TCO and ultraviolet radiation (photolysis rate of the reaction of $O_3$ to $O(^1D)$ or $JO^1D$) near the Greek towns of Kastelorizo (36° N) with ionosonde total electron content (TEC) measurements during the solar eclipse of 29 March 2006. The TCO was measured with a Brewer spectrophotometer and dipped near the time of totality by 75 DU (or 25 %), allegedly due to contamination by diffuse sky radiance in the instrument's field of view as mentioned above. To remove this artifact, two second-order polynomials were fitted to the data points and residual to this fit were calculated. Residuals had an amplitude of about 5–10 DU (1.7–3.5 %). Spectral Fourier analysis revealed oscillations in TCO with a dominant period in the range of 28–38 minutes and a secondary oscillation with a period of 12–13 minutes. Cross-Spectrum analysis of $JO^1D$ against ionospheric TEC revealed a distinct covariance between the frequency components of both parameters, suggesting that oscillations in TEC and TCO were driven by gravity waves initiated in the stratosphere.

Lastly, Bernhard and Petkov (2019) (hereinafter B&P19) reported increases in TCO by ~ 8 % during the "Great American Eclipse" of 21 August 2017, which were symmetrical about totality. However, the increase disappeared after correcting for the LD effect (Sect. 1.2), suggesting that it is not caused by changes in TCO. Furthermore, oscillations in TCO before and after totality were not observed.

Since several authors attribute the observed changes to the solar LD effect and gravity waves, we assess both phenomena in the following.

## 1.2 Solar limb darkening

The TCO is typically derived from measurements at one or two wavelengths affected by ozone (e.g., 305 nm) and one reference wavelength not affected by ozone (e.g., 340 nm). Standard ozone retrieval algorithms assume that the Sun's





spectrum outside the Earth's atmosphere is constant at these wavelengths. This assumption breaks down during a solar eclipse because the disk of the Sun is not uniformly bright. The temperature of the Sun's photosphere decreases with increasing distance from the Sun's center and photons emanating near the solar limb originate from a shallower (and cooler) layer of the photosphere than photons from the center of the solar disk. As a result, the solar limb appears darker than the
Sun's center. This effect is known as solar limb darkening (LD). Shorter wavelengths are more strongly affected than longer wavelengths. Hence, the ratio of solar spectral irradiances at 305 and 340 nm outside the Earth's atmosphere is smaller during a solar eclipse, when the center of Sun is concealed, compared to a "normal" day, when the Sun is unobstructed. If the ozone retrieval algorithm does not consider a LD correction, the smaller spectral irradiance at 305 nm is incorrectly interpreted as a larger ozone column.

Figure 1 shows as an example of the LD correction for measurements of TCO during the total solar eclipse of 21 August 2017 (B&P19). For the uncorrected dataset, TCO exhibits an unrealistic increase by about 23 DU or 8 %. The LD correction by Waldmeier (1941) decreases this peak only marginally. When corrected for the LD effect using either the parameterizations by Pierce and Slaughter (1977) or Neckel (2005), the artifact is reduced to ± 2.6 DU (± 0.9 %). Some of the remaining variation (e.g., the gradual increase before totality) could be caused by actual changes in TCO. Between 17:36
and 18:36, LD-corrected TCO data decreased by 1 DU (0.33 %) with no obvious oscillations. Between 18:38 and the end of the measurements at 19:32, TCO remained constant to within 0.3 DU (0.10 %).

Several recent papers (e.g., Blumthaler et al., 2006; Emde and Mayer, 2007; Kazadzis et al., 2007; Kazantzidis et al., 2007) have used the LD parameterization by Waldmeier (1941), which does not capture the full magnitude of the LD effect. This would explain why peaks in TCO near totality are still apparent in these publications after LD correction. We suspect
that the LD correction applied by Bojkov (1968) discussed above was also too low.

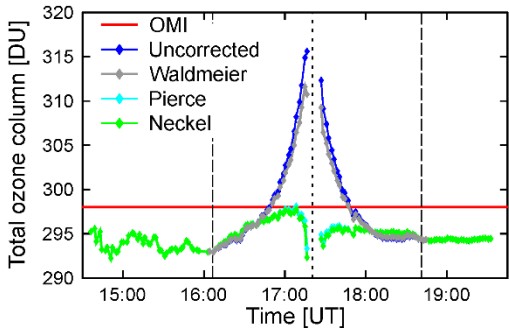

Figure 1: Total column ozone measured by a GUVis-3511 multi-filter radiometer during the total solar eclipse of 21 August 2017. Uncorrected measurements are compared with datasets corrected using the LD parameterizations by Waldmeier (1941), Neckel (2005), and Pierce and Slaughter (1977). Dashed and broken lines indicate the period and time of maximum eclipse, respectively. Reproduced from B&P19, which has been published under the Creative Commons Attribution 4.0 License.



### 1.3 Potential effects of gravity waves on total column ozone

During a solar eclipse, the Moon's shadow (umbra and penumbra) moves at supersonic speed over the Earth's atmosphere. Based on modeling, Chimonas (1970), Chimonas and Hines (1970, 1971), and Chimonas (1973) hypothesized that the resulting cooling of the atmosphere of up to a few degrees Celsius will generate atmospheric gravity waves, which may either originate in the troposphere through cooling of water vapor; in the stratosphere, by the cooling of the ozone layer; and in the thermosphere, where variations in temperature are most pronounced. The gravity waves are generated as an adjustment after a sudden change in forcing has caused the atmosphere to depart from its large-scale balanced state (Williams et al., 2003; Marlton et al., 2016). Because the shadow of the Moon is moving faster than the characteristic speed of waves in air, gravity waves manifest themselves as three-dimensional (3-D) bow waves that are analogous to surface waves that form in the wake of a ship. Bow waves may propagate vertically and horizontally, and may be detectable as a ground-level atmospheric pressure variation or a travelling ionospheric disturbance (Seykora et al., 1985).

A number of investigators have found evidence of eclipse-driven pressure changes at the surface using sensitive microbarometers, which can resolve pressure variations at the 1 Pa (0.01 mbar) level as summarized by Farges et al. (2003), Marty et al. (2013) and Marlton et al. (2016), and references therein. Observed pressure perturbations were generally between 0.1 Pa (Goodwin and Hobson, 1978) and 25 Pa (Anderson et al., 1972), and had periods ranging between 15 minutes and four hours. One conclusion from these studies is that perturbations can only be reliably attributed to gravity waves if weather conditions at the time of these observations were stable. In many studies, large pressure fluctuations associated with local changes in weather prevented the detection of eclipse-induced pressure perturbations (e.g., Anderson and Keefer (1975)). Furthermore, the magnitude and period of pressure perturbations cannot be clearly linked to modeling proposed in theoretical studies (Chimonas, 1970; Fritts and Luo, 1993; Eckermann et al., 2007). We operated a microradiometer at Mazatlán (Sect. 3.4) attempting to find evidence for bow waves at the surface that could explain potential variations in TCO.

Because most observations during solar eclipses are hampered by less-than-ideal observing conditions, solid proof for eclipse-generated gravity waves was still lacking as of 2017, more than 45 years after the effect was first proposed by Chimonas (1970). This changed in late 2017 when Zhang et al. (2017) reported on high-fidelity, wide-coverage ionospheric observations, which were taken during the "Great American Eclipse" of 21 August 2017 using data from Global Navigation Satellite System (GNSS) receivers at ~ 2,000 sites distributed across North America. Because of the high density of receiver stations, maps of the ionospheric TEC could be produced during the progression of the eclipse. The results indicate that the wave's ripples were indeed bow shaped, traveling at ~ 280 m/s with a directional azimuth of ~ 120°. These results were in excellent agreement with the theoretical predictions of Chimonas (1970) and provided compelling observational evidence to support the general mechanism of bow wave excitation by the supersonic shadow of the Moon.

Following these encouraging results, Colligan et al. (2020) confirmed that eclipse-induced bow waves also exist in the stratosphere using radiosonde data collected at a field campaign in Chile during the total solar eclipse of 2 July 2019.



According to theory, a radiosonde subjected to the influence of a gravity wave will experience an elliptical motion. Such motion was indeed observed on three instances. Colligan et al. (2020) concluded that these measurements represent the first unambiguous detection of eclipse-induced gravity waves in the middle atmosphere at about 25 km.

## 1.4 Day/night differences in total column ozone

Day/night differences in TCO have recently been quantified at Bern (47.0° N) and Payerne (46.8° N), Switzerland, using
ground-based ozone microwave radiometers and several model-based datasets (Sauvageat et al., 2023). For June (Figure 3c by Sauvageat et al. (2023)), all dataset show a consistent day/night cycle with virtually no difference below 20 km, higher ozone concentrations during the day than night between 20 and 45 km, with a peak difference of 4 % at ~ 40 km, followed by a large daytime ozone depletion at altitudes larger than 45 km with maximum differences ranging between –62 % and –76 % at ~ 72 km. The large decline of ozone concentrations in the mesosphere during daytime is caused by photochemical
reactions initiated by sunlight (Dikty et al., 2010). Considering that about 90 % of ozone is at altitudes below 35 km, the day/night cycle should have little effect on TCO. Furthermore, positive differences in the upper stratosphere partially cancel mesospheric declines. These data are qualitatively consistent with similar measurements collected by Parrish et al. (2014) at the Mauna Loa observatory (19.5° S), although the maximum enhancement in the upper stratosphere at this location is only ~ 2 % and the mesospheric daytime depletion is also somewhat smaller than that over Switzerland. Using the Air Force
Geophysical Laboratory (AFGL) atmospheric constituent profile for mid-latitude summer (Anderson et al., 1986) and the work by Sauvageat et al. (2023), we calculated that the TCO is higher by 0.6 % during the day. It can therefore be expected that TCO variations during the relatively short period of a solar eclipse remain below 0.6 %. However, Sauvageat et al. (2023) show that the transition from the night to the day regime and vice versa occurs within a short period, which is comparable to the duration of a solar eclipse. Our estimate of day/night TCO variations is consistent with the conclusion by
Chakrabarty et al. (1997) that "short-term fluctuations in TCO during a solar eclipse are not expected due to the long lifetime of ozone in the stratosphere". However, momentum from gravity waves; the rapid cooling of the atmosphere during a solar eclipse, which is considerably faster than the day/night cycle; and a change in tropopause height (Dutta et al., 2011), which could affect the ozone profile, could potentially lead to a somewhat larger effect on the TCO. Discussions of the reasons for the day/night cycle in stratospheric and mesospheric ozone are beyond the scope of this paper but are provided in the work
by Natarajan et al. (2023).

## 2 Eclipse parameters, location and local conditions

Table 1 compares basic eclipse parameters for the eclipses at Mazatlán, Fort Collins and Union Glacier provided by the Astronomical Applications Department of the U.S. Naval Observatory (USNO) at https://aa.usno.navy.mil/data/SolarEclipses. Our calculations (Sect. 4.2) of the times of 1st and 4th contact and eclipse
maximum are also indicated. These times agree within a few seconds with USNO data. All times in the table and the



remainder of the paper refer to Coordinated Universal Time (UTC). To convert from UTC to local time, subtract 7, 6, and 3 hours from the time at Mazatlán, Fort Collins, and Union Glacier, respectively.

Table 1. Comparison of Eclipse calculations by us and USNO.

| | Time (UTC[a]) | | Difference | Solar zenith | Solar Azimuth |
|---|---|---|---|---|---|
| | Our calculation | USNO | [seconds] | [°] | [°] |
| *Mazatlán, Mexico, 8 April 2024 (23.1836° N, 106.4254° W), magnitude: 1.022* | | | | | |
| Start partial eclipse (1st contact) | 16:51:20 | 16:51:16 | +4 | 36.2 | 110.0 |
| Start total eclipse (2nd contact) | | 18:07:15 | | 21.2 | 134.8 |
| Maximum eclipse | 18:09:30 | 18:09:25 | +5 | 20.9 | 136.0 |
| End total eclipse (3rd contact) | | 18:11:36 | | 20.5 | 137.2 |
| End partial eclipse (4th contact) | 19:32:03 | 19:31:58 | +5 | 16.7 | 201.8 |
| *Fort Collins, Colorado, 14 October 2023 (40.5704° N, 105.0954° W), magnitude: 0.829* | | | | | |
| Start partial eclipse (1st contact) | 15:13:55 | 15:14:01 | −6 | 69.0 | 122.0 |
| Maximum eclipse | 16:35:31 | 16:35:34 | −3 | 57.3 | 140.6 |
| End partial eclipse (4th contact) | 18:04:50 | 18:04:43 | +7 | 49.8 | 166.4 |
| *Union Glacier, Antarctica, 4 December 2021 (79.7594° S, 82.8381° W), magnitude: 1.002* | | | | | |
| Start partial eclipse (1st contact) | 06:53:47 | 06:53:46 | +1 | 77.2 | 158.2 |
| Start total eclipse (2nd contact) | | 07:44:54 | | 76.1 | 146.1 |
| Maximum eclipse | 07:45:26 | 07:45:17 | +9 | 76.1 | 146 |
| End total eclipse (3rd contact) | | 07:45:42 | | 76.1 | 145.9 |
| End partial eclipse (4th contact) | 08:37:17 | 08:37:15 | +2 | 74.6 | 133.7 |

[a]To convert from UTC to local time, subtract 7, 6, and 3 hours from the time at Mazatlán, Fort Collins, and Union Glacier, respectively.


## 2.1 Mazatlán, Mexico

Observations at Mazatlán were performed during the total solar eclipse on 8 April 2024 at the Institute of Marine Sciences and Limnology (Instituto de Ciencias del Mar y Limnología) of the National Autonomous University of Mexico (Universidad Nacional Autónoma de México (UNAM)). The GUVis-3511 radiometer with serial number 361 was set up on

the tallest building of the institute (Figure 2) at a latitude of 23.18360° N, a longitude of 106.42543° W, and 19 ± 2 m above sea level. In approximately NW direction of the instrument is a radio tower, which is about 6.5 m higher than the instrument's collector and 2.75 m away, resulting in an angular height of 67°. The angular extent in azimuth is 5.8°. However, the tower only obstructs about 20 % of this angular range because of its lattice structure (Figure 2c). Assuming isotropic sky light, we determined that the restriction of the horizon by mountains and other landscape features reduces the

horizontal irradiance by 0.23 %. Shading of the sky by the tower leads to an additional reduction by 0.27 %. Hence, the total reduction by features above the horizon is 0.50 %, which is well below the uncertainty of the measurements. Measurements were also performed on 7 and 9 April 2024 to compare with data collected on eclipse day. Thin cirrus clouds were present on



the day before the eclipse and the day of the eclipse (Figure 2d), which affected the measurements as explained in Sect. 5.2.
The sky on the day after the eclipse was free of clouds and this day serves as a reference day against which measurements on
eclipse day are compared.

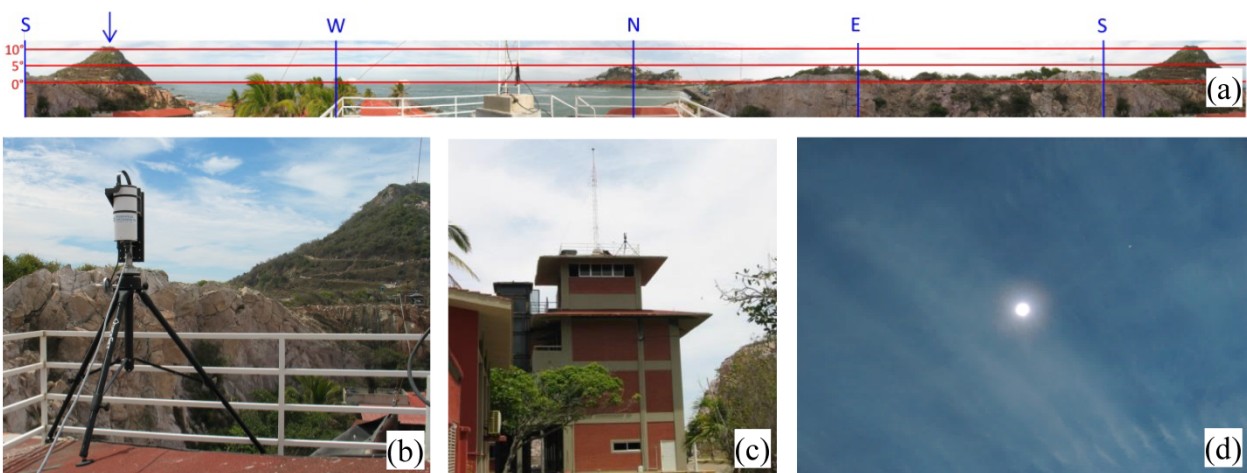

Figure 2: Measurement site at Mazatlán. (a) Panorama as seen from the instrument's location. The highest point, indicated by an arrow in
SSW direction, is the lighthouse of Mazatlán ("Faro Mazatlán"), which is 681 m away from the instrument and extends 11.5° above the
horizon. In approximately NW direction is a radio tower, which is about 6.5 m higher than the instrument's collector and 2.75 m away,
resulting in an angular height of 67°. Other obstructions are less than 5° above the horizon. (b) GUVis-3511 instrument set up on the roof
of the institute. (c) View towards N with GUVis-3511 on the right side of the radio tower on the institute's roof. (d) Thin cirrus clouds
during totality. The Sun's corona is overexposed to emphasize the sky condition.

## 2.2  Fort Collins, Colorado

Observations at Fort Collins were performed at the headquarters of the USDA UV-B Monitoring and Research Program at a
latitude of 40.5704° N, a longitude of 105.0954° W, and 1,536 m above sea level with the same GUVis-3511 radiometer that
was used in Mazatlán. The site is adjacent to the Fort Collins campus of the Colorado State University. Measurements were
taken during the annular solar eclipse of 14 October 2023. However, Fort Collins was not under the path of annularity; hence
only a partial eclipse was observed. The irradiance at the time of maximum eclipse was reduced to 21.3 % of the uneclipsed
Sun. As the instrument was set up near ground level, measurements were affected by obstructions from nearby buildings and
trees; however, the Sun was not obstructed during the entire time of observations. The sky was free of clouds until 17:15
with thin cirrus clouds appearing thereafter. Thus, measurements were not affected by clouds between the 1[st] contact, and
past the time of maximum solar obscuration at 16:35:34.



## 2.3 Union Glacier, Antarctica

Observations at Union Glacier were performed at the Union Glacier Joint Scientific Polar Station (Estación Polar Científica Conjunta Glaciar Unión), which is operated jointly by the Chilean Antarctic Institute (INACH) and the Chilean armed forces at a latitude of 79.7594° S, a longitude of 82.8381° W, and 765 m above sea level using a GUVis-3511 radiometer with serial number 401. Additional ground-based spectral instruments included a sunphotometer affiliated with NASA's Aerosol Robotic Network (AERONET); Sect. 3.3. Measurements of this instrument showed that the aerosol optical depth (AOD) was generally below 0.025 at Union Glacier during the eclipse, confirming the pristine conditions at deep field locations in Antarctica. There were ideal observation conditions throughout the day of the eclipse and the following day with no clouds. However, the solar elevation at totality was only 14°. This low elevation makes the interpretation of data more challenging compared to the conditions at the other sites because observations of TCO become more sensitive to the vertical distribution of ozone in the atmosphere (Ockenfuß et al., 2020). Furthermore, the high surface of albedo of 0.95 at this site (Cordero et al., 2014) greatly enhances multiple scattering between the surface and the overlying atmosphere, thus complicating the radiative transfer further.

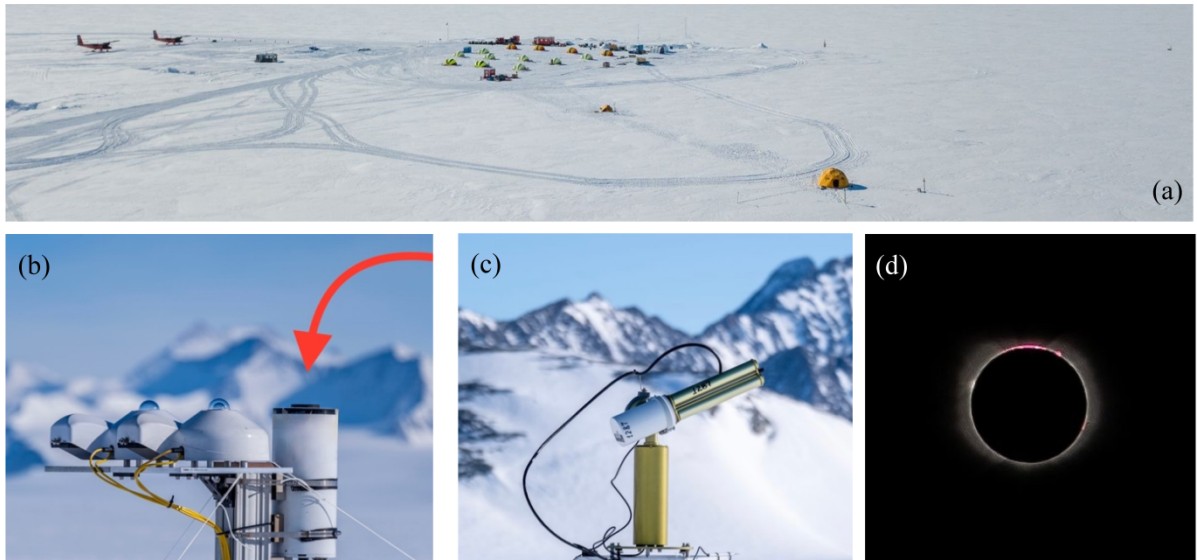

Figure 3: Measurement site at Union Glacier. (a) Union Glacier Joint Scientific Polar Station. The GUVis-3511 radiometer was set up close to the yellow tent in the foreground. (b) GUVis-3511 radiometer (red arrow) with pyranometers in the foreground. (c) CE-318 sunphotometer at Union Glacier, which is part of NASA's AERONET network. (d) View of the Sun during totality. Several prominences, Baily's Beads, and the Sun's chromosphere and inner corona are also visible.





## 3 Instrumentation and calibration

### 3.1 GUVis-3511 radiometers

Measurements at Mazatlán and Fort Collins were performed with the same GUVis-3511 (hereinafter GUV) multi-channel filter radiometer from Biospherical Instruments, Inc. (BSI) that was used to observe the 2017 total solar eclipse in Oregon as described by B&P19. In brief, the instrument is equipped with 18 channels with spectral bandwidths of approximately 10 nm and the following nominal wavelengths (nm): 305, 305, 313, 320, 340, 340, 380, 395, 412, 443, 490, 532, 555, 555, 665, 875, 940, and 1020. Data from a 19th channel measuring photosynthetic active radiation (PAR) were not used in this study.

Note that the instrument has duplicate channels at 305, 340, and 555 nm, which are equipped with either a standard production photodiode or an alternative photodiode. As measurements of the two types proved to be very consistent (B&P19), we only use measurements of the standard photodiode here.

The radiometer was equipped with a computer-controlled shadow band known as BioSHADE to provide measurements of direct and global (Sun and sky) spectral irradiance (Hooker et al., 2012; Witthuhn et al., 2017). At Mazatlán, the band

executed a shadow band "sweep" every three minutes. Each 3-minute period resulted in about 90 seconds of global spectral irradiance measurements (with the shadow band stowed while sampling at 1 Hz) and 66 seconds of data sampled at 15 Hz during which the band rotated by 180°. At Fort Collins, the BioSHADE was programmed to perform a sweep every 2 minutes, resulting in 30 seconds of global spectral irradiance measurements at 1 Hz followed by 70 second long sweeps while sampling at 15 Hz to adequately resolve the short period when the band's shadow moves over the instrument's

diffuser. Because there were some problems with the band's operation, shadow band sweeps were not available before 15:35 and sweeps after 17:51 occurred at somewhat irregular intervals, ranging between 90 and 160 seconds. These irregularities have little effect on the results described below. A detailed description of the instrument's optical and electrical specifications, spectral response functions, and cosine error can also be found in the work by B&P19.

Channels between 305 and 380 nm of the GUV used in Mazatlán and Fort Collins were vicariously calibrated against

"Version 2" (Bernhard et al., 2004) measurements of the SUV-100 spectroradiometer located on the roof platform of BSI using data collected on the cloudless day of 16 April 2024. Version 2 data have a spectral resolution of 1 nm full width at half maximum (FWHM). All other channels were calibrated against a lamp traceable to NIST standard F-616, which was calibrated by the U.S. National Institute of Standards and Technology (NIST) against the scale of spectral irradiance established by NIST in 2000 (Yoon et al., 2002). The methods of the vicarious and lamp-based calibration transfers are

described in more detail in the supplement of B&P19.

The instrument used at Union Glacier was also a GUVis-3511 but did not have a shadow band. It was equipped with channels at the following wavelengths (nm): 305, 313, 340, 380, 412, 555, 670, 875, 1020, 1245, and 1640. All channels



were calibrated at BSI with a NIST-traceable standard in September 2015 and channels between 305 and 380 nm were also vicariously calibrated against SUV-100 Version 2 data using measurements collected between 26 August and 2 September

2015. The instrument was recalibrated at the University of Santiago, Chile, on 18 March 2022 using a Bentham Instruments CL6 spectral irradiance standard traceable to the Physikalisch-Technische Bundesanstalt (PTB) in Germany. The two lamp-based calibrations agreed to within 3.5 % at all wavelengths but the 2022 calibration was considered more reliable because of the closer proximity in time with the measurements at Union Glacier. The calibration used for processing solar data at wavelengths between 305 and 380 nm was the vicarious calibration of 2015 scaled by the ratio of the lamp-based

calibrations of these channels in 2022 and 2015, while the 2022 lamp calibration was used for all other channels. The spectral and angular response functions were not characterized. We therefore used generic response functions of this instrument model. The missing angular response data led to increased uncertainty in the cosine error correction and some artifacts in the data as discussed in Sect. 5.4.1.

### 3.2 Microtops II Ozonometer

A Model 521 Microtops II® Ozonometer from Solar Light was used at Mazatlán to provide measurements of TCO independent from the GUV instrument. The instrument is a sun photometer and derives TCO from measurements of direct solar irradiance at 305, 312, and 320 nm. The bandwidth of each filter is 2.4 nm. Data processing is described by Morys et al. (2001). The instrument was purchased shortly before the eclipse and had a current factory calibration from Solar Light. Measurements of TCO by the instrument were consistently higher by ~ 6.5 % relative to observations of the GUV and the

Ozone Monitoring Instrument (OMI) on NASA's Aura satellite, which were downloaded from NASA's Giovanni data server at https://giovanni.gsfc.nasa.gov. We scaled measurements down by this amount to better facilitate comparisons with GUV data. The instrument is manually pointed at the Sun and measurements are therefore sensitive to pointing errors. To reduce the associated uncertainty, three to five measurements were taken consecutively and their average was calculated. Averages were only used if the standard deviation calculated from the individual measurements was less than 3 DU.

### 265 3.3 AERONET sunphotometer

The GUVis-3511 at Union Glacier was collocated with a CE-318 sunphotometer from CIMEL Electronique, which is part of NASA's Aerosol Robotic Network (AERONET) (Holben et al., 1998). The instrument measures AOD at 340, 380, 440, 500, 675, 870, 1020, and 1640 nm. Version 3 (Sinyuk et al., 2020), Level 1.0 AOD data were downloaded from the AERONET website at https://aeronet.gsfc.nasa.gov/. These data are not clouds screened but AODs measured before the 1[st] and after the

4[th] contact are identical with Level 1.5 data, which are cloud screened on quality controlled. (We used Level 1.0 data because the period of the eclipse is excluded from the Level 1.5 dataset.) Both Level 1.0 and Level 1.5 may not have the final calibration applied because the instrument has not yet been returned from Antarctica for a post-deployment calibration. The uncertainty of AERONET field data is in the range of 0.01–0.021 (Sinyuk et al., 2020; Eck et al., 1999).





### 3.4 Microbarometer

We operated a microbarometer (model PTB210 BAROCAP Digital Barometer from Vaisala) at Mazatlán in an attempt to observe gravity waves that may lead to small changes in pressure at the surface. It has a resolution of 0.01 hPa and was operated at 1 Hz and with a model SPH10 static pressure head from Vaisala, which reduces the effect of wind on pressure measurements. The instrument was purchased shortly before the eclipse and had a factory calibration from Vaisala.

### 4   Data processing

Measurements at Mazatlán and Fort Collins were processed using the software package "GUVis-3511 Data Processor" (hereinafter GUVDP), which was developed in 2020 and is based on the methods described in the supplement of B&P19. The software and its theoretical background are described in a manual (see Section "Code and data availability"). In brief, GUVDP ingests uncalibrated raw data from the GUV radiometer and applies calibration functions that depend on the solar zenith angle (SZA) and TCO occurring at the time of the measurement. If the GUV radiometer is equipped with a

BioSHADE, GUVDP also produces several secondary data products including but not limited to the TCO, the cosine-error corrected global irradiance, the ratio of direct and global irradiance, and the AOD. Furthermore, GUVDP performs Langley analyses, which can be used to for the calibration of AOD measurements. Calibrated measurements report the solar spectral irradiance at a spectral resolution of 1 nm. More specifically, calibrated measurement resemble measurements of a hypothetical spectroradiometer with a slit function $s(\lambda)$, where $s(\lambda)$ is a triangular function with a bandwidth of 1 nm

FWHM. Version 2 SUV-100 data are also normalized to this slit function.

GUVDP software currently does not support processing of data where the vicarious calibration was executed at a location with low albedo (e.g., San Diego) and field measurements are performed at a location with high albedo (e.g., Union Glacier). The software also does not apply a cosine error correction if shadowband data are not available. For these reasons, measurements of the GUV deployed at Union Glacier were processed manually but using the same methods as implemented

in GUVDP software. The ratio of direct to global irradiance, which is required for the cosine error correction, was calculated with the radiative transfer model UVSPEC (Mayer and Kylling, 2005) in lieu of deriving this ratio from shadow band data.

### 4.1   Calculation of total column ozone

Total column ozone was calculated with GUVDP software from GUV measurements of *global* irradiance with lookup tables

based on a method proposed by Stamnes et al. (1991). These lookup tables are derived from spectra calculated with UVSPEC, which are convolved with a triangular function of 1 nm FWHM. To derive TCO for the 340 nm / 305 nm wavelength pair, ratios named $Q_{340/305}(\theta, \Omega)$ of spectral irradiance at 340 nm (a wavelength weakly absorbed by ozone) and at 305 nm (a wavelength strongly absorbed by ozone) are calculated as a function of SZA (symbol $\theta$) in steps of 1° and





TCO (symbol $\Omega$) in steps of 20 DU. The resulting lookup table is spline-interpolated to SZA steps of 0.1° because early

results showed that linear interpolation will lead to small (< 0.5 %) variations in TCO with a periodicity of 1° in SZA, which

could be falsely interpreted as changes in TCO during an eclipse. This lookup table is then multiplied with a correction term

$K_{340/305}(\theta,\Omega,\theta_c,\Omega_c)$ to adjust for the relatively broad spectral response functions of the GUV, resulting in a modified

lookup table $\widetilde{Q}_{340/305}(\theta,\Omega)$:

$$\widetilde{Q}_{340/305}(\theta,\Omega) = Q_{340/305}(\theta,\Omega) \times K_{340/305}(\theta,\Omega,\theta_c,\Omega_c) = Q_{340/305}(\theta,\Omega)\frac{C_{340}(\theta,\Omega)/C_{340}(\theta_c,\Omega_c)}{C_{305}(\theta,\Omega)/C_{305}(\theta_c,\Omega_c)}. \qquad (1)$$

The arguments $\theta_c$ and $\Omega_c$ indicate the SZA and TCO at the time of calibration. $C_i(\theta,\Omega)$, with $i = 340$ or $305$, is defined as:

$$C_i(\theta,\Omega) = \frac{\int E_m(\lambda_i,\theta,\Omega)r_i d\lambda}{E_m(\lambda_i,\theta,\Omega)}, \qquad (2)$$

where $E_m(\lambda_i,\theta,\Omega)$ is the modeled spectral irradiance at wavelength $\lambda_i$ and $r_i$ is the spectral response function of channel $i$. Note that the correction term $K_{340/305}(\theta,\Omega,\theta_c,\Omega_c)$ is equal to one if $\theta = \theta_c$ and $\Omega = \Omega_c$. TCO is finally calculated from the modified lookup table $\widetilde{Q}_{340/305}(\theta,\Omega)$, the SZA $\theta$ at the time of the solar observations, and the ratio $P_{340/305}(\theta)$ of

calibrated measurements at 340 and 305 nm, defined as:

$$P_{340/305}(\theta) = \frac{V_{340}(\theta)/R_{340}(\theta_c,\Omega_c)}{V_{305}(\theta)/R_{305}(\theta_c,\Omega_c)}, \qquad (3)$$

where $V_{340}(\theta)$ and $V_{305}(\theta)$ are the uncalibrated measurements of the GUV at 340 and 305 nm, and $R_{340}(\theta_c,\Omega_c)$ and

$R_{305}(\theta_c,\Omega_c)$ are the responsivities established during calibration at $\theta_c$ and $\Omega_c$.

TCO is also calculated from the wavelength pair of 340/313 nm using a similar approach. In the following, TCO data

calculated from the 340/305 and the 340/313 pair are referred to as TCO$_{340/305}$ and TCO$_{340/313}$, respectively. All ozone data are available at a frequency of 1 Hz during times when the shadow band is inactive.

## 4.2 Solar limb darkening correction

The solar LD correction was calculated analogous to that for the 2017 total solar eclipse described by B&P19. In brief, elevation, azimuth, and angular radii of Sun and Moon were downloaded from the Horizons System of the Jet Propulsion

Laboratory (JPL; https://ssd.jpl.nasa.gov/?horizons) and the fraction of the solar disk as function of time during the progression of the eclipse was calculated with the algorithm by Koepke et al. (2001). However, we replaced the parameterization of the wavelength dependence of solar LD used by Koepke et al. (2001), which is based on the work by Waldmeier (1941), with the parameterization by Pierce and Slaughter (1977) and Pierce et al. (1977), which is based on data collected by the McMath-Pierce Solar Telescope of the National Solar Observatory on Kitt Peak. Figure 4 shows results of



the calculations for Mazatlán. Between about 17:30 and 19:00, the shortest wavelength (305 nm) is attenuated the most while the longest wavelength (1020 nm) is attenuated the least.

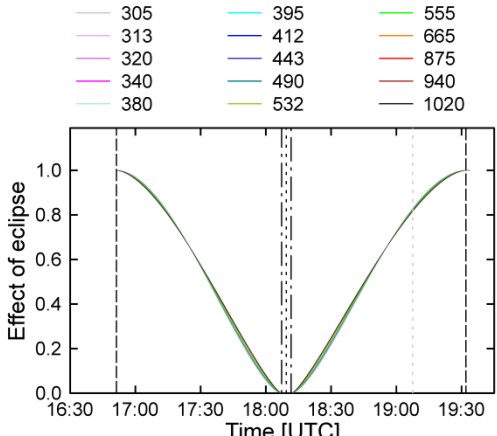

Figure 4: Change in the extraterrestrial irradiance as a function of time at Mazatlán, calculated from JPL ephemeris data and the parameterizations of solar limb darkening published by Pierce and Slaughter (1977) and Pierce et al. (1977) for the wavelengths of the GUV radiometer. Vertical long-dashed lines indicate the times of the 1$^{st}$ and 4$^{th}$ contact, the dashed-dotted lines indicate the times of the 2$^{nd}$ and 3$^{rd}$ contact, the short-dashed line indicates the time of the eclipse maximum, and the gray dashed line indicates local solar noon.

## 5  Results

### 5.1  Comparison of total column ozone measurements in the San Diego

After the GUV was calibrated, TCO measured at BSI's headquarter in San Diego for the period of 15–19 April 2024 was calculated from data of the GUV radiometer and compared with TCO data of the collocated SUV-100 spectroradiometer. The latter were calculated from Version 2 spectra based on the method described by Bernhard et al. (2003). This method also uses measurements of global irradiance; however, it does not use lookup tables. Instead, measured spectra are compared with UVSPEC spectra calculated with different TCO as input, and the TCO of the model spectrum that leads to the best

agreement between measurement and model is the TCO returned by the algorithm. The method was mainly developed to retrieve TCO from global irradiance at high latitudes where large SZAs are prevailing and where TCO becomes critically depend on the vertical distribution of ozone in the atmosphere. Hence, the method allows to change profiles on a daily basis.

   Figure 5a shows measurements of the two instruments plus observations by OMI and the Microtops. The sky was free of clouds until local solar noon of 17 April 2024. Measurements on the last day were affected by clouds as indicated by

measurements of spectral irradiance at 340 nm (Figure 5b). For the clear-sky period, TCO data derived from the GUV are biased low by −1.1 % relative to the SUV-100 data, both for TCO$_{340/305}$ and TCO$_{340/313}$. The standard deviation of the



difference is 0.6 % for $TCO_{340/305}$ and 1.0 % for $TCO_{340/313}$. For the cloudy period, the bias is similar but the standard deviation is increased to 2.4 % and 3.5 %, respectively. The TCO measured by OMI is about 2 % lower than GUV data while measurements by the Microtops are higher by 6.5 % as mentioned in Sect. 3.2.

Results shown in Figure 5a confirm that TCO measurements by the GUV are accurate to within 2–3 % and capture variations in TCO (such as the drop by ~ 20 DU on 16 April 2024) well. $TCO_{340/313}$ is more affected by clouds than $TCO_{340/305}$ because measurements at 305 nm are more strongly affected by ozone than those at 313 nm.

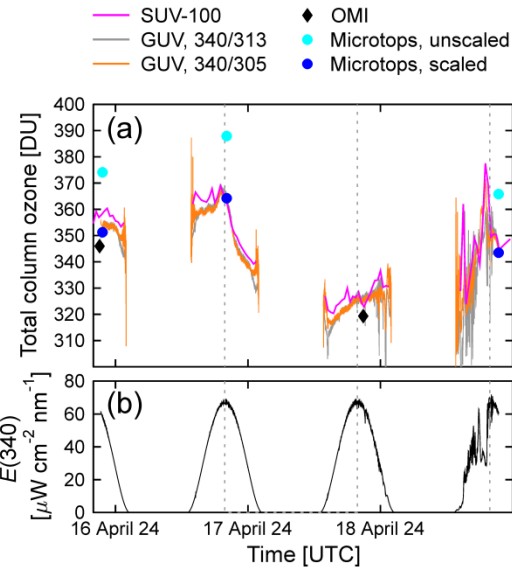

Figure 5: (a) Total column ozone at San Diego calculated from Version 2 spectra of the SUV-100 spectroradiometer (pink line), derived from GUV measurements for the 340/313 (gray line) and 340/305 (orange line) wavelength pairs, provided by OMI (black diamond), and measured by the Microtops (cyan circle). Microtops measurements scaled by 1/1.065 are also shown. Only data up to SZAs of 85° are included. (b) Spectral irradiance at 340 nm measured by the GUV radiometer. Vertical broken lines indicate the times of local solar noon.

## 5.2  Measurements at Mazatlán

### 5.2.1    Spectral irradiance

Figure 6a shows global spectral irradiance measured by the GUV-3511 radiometer at Mazatlán on 8 and 9 April 2024 plus model calculations with input parameters optimized for 8 April. Specifically, TCO was set to 283 DU (TCO retrieved by the GUV close to totality) and AOD was parameterized with Ångström's turbidity formula by setting the Ångström coefficients α and β to α=1.1080 and β=0.0234. These parameters were determined by fits to AOD data derived with GUVDP software

(Sect. 5.2.2). Figure 6b shows the ratio of measurements on 8 April 2024 and the model. Lastly, Figure 6c shows the ratio of measurements corrected for the LD effect (i.e., the LD functions plotted in Figure 4) and the model. Specifically, raw data





were divided the LD functions and then reprocessed with GUVDP. It can be seen that this correction removes the effect of the eclipse with high fidelity.

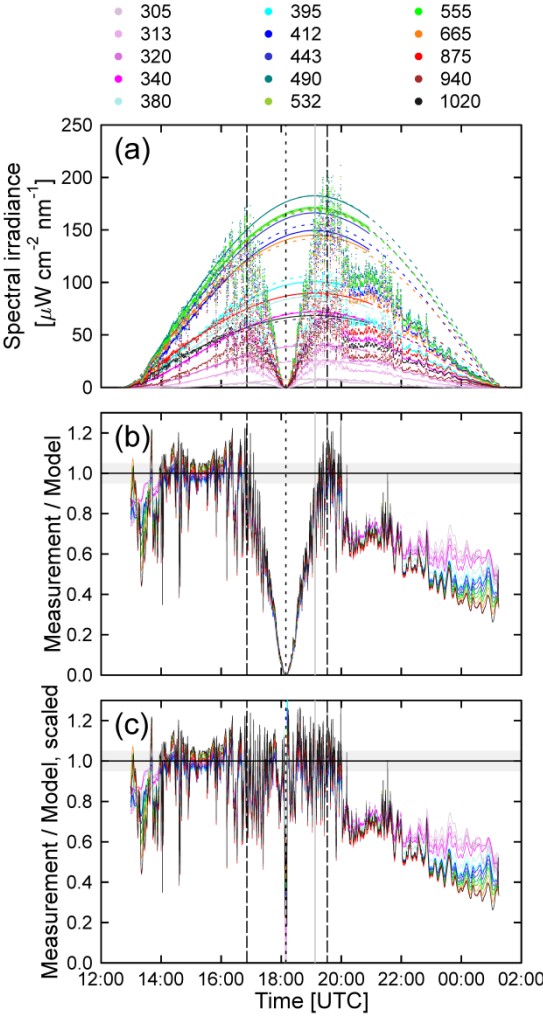

Figure 6: (a) Global spectral irradiance measured on 8 April 2024 (small dots) and on 9 April 2024 (solid smooth lines), and the modeled spectral irradiance with input parameters optimized for 8 April 2024 (broken lines). (b) Ratio of measurement and model for 8 April 2024. The gray band indicates the region of ± 5 % about the ideal ratio of one. Ratios for 940 nm are not shown because of the uncertainty due to water vapor absorption. (c) Same as (b), but measured data were divided by the LD function shown in Figure 4. Long-dashed lines in all panels indicate the 1st and 4th contact. The time of the eclipse maximum is indicated by a short-dashed line and the local solar noon is indicated by a gray line.


Figure 6 allows the following conclusions:

- Measurements on 8 April 2024 were affected by clouds, which resulted in high variability of the measured spectral irradiance. Comparison with the model indicates that broken cirrus clouds (Figure 2d) occurring between 14:00 and 20:00 (a period including the eclipse) mostly added scatter about unity. During this period, measurements varied by

approximately ± 25 % about the value expected for clear skies. This indicates that broken cirrus clouds above the observation site both reduced the irradiance (when a cloud obscured the solar disk) or enhanced it (when the solar disk was free and additional radiation reached the detector from bright white clouds surrounding the Sun). The



cloud effect was the least between 15:15 and 15:30 when the ratio of measurement and model was within ± 5 % of one at all wavelengths.

• After 20:00, broken cirrus cloud turned into overcast conditions. Later in the day, longer wavelengths were more strongly attenuated than shorter ones, as one would expect for a thick cloud.

• The sky was free of clouds on 9 April throughout the day and measurements on this day agree well with the model, giving confidence in both the measurements and model calculations.

Figure 7 shows a close-up of the spectral irradiance within ± 10 minutes of totality. As was observed in 2017, the spectral irradiance decreases by more than two orders of magnitude when transitioning into totality. The spectral irradiance during totality is not constant and is lowest at approximately the time of the eclipse maximum. At this time, the instrument performed a shadow band sweep, but this is not obvious in the data. In fact, the variability due to moving cirrus clouds was likely larger than the shading effect of the band.


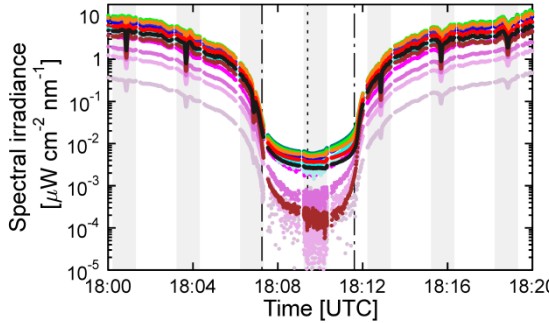

Figure 7: Close up of change in spectral irradiance within ± 10 minutes of totality. Before 18:03 and after 18:13, data are based on cosine error corrected data. Close to totality, the cosine error correction becomes uncertain and data were scaled by the diffuse correction factor assuming isotropic sky radiance. The color coding is identical with that used in Figure 6. Periods of shadow-banding are indicated by gray shading. The times of the 2nd and 3rd contact are indicated by dashed-dotted lines and the time of maximum eclipse is indicated by a short-dashed line.

### 5.2.2 Aerosol optical depth

Figure 8 shows AOD at Mazatlán derived with GUVDP software. All AOD measurements on 8 April were affected by clouds. The AOD measured on 16:07 was the lowest AOD of that day and the least affected by clouds. AODs for the cloudless day of 9 April were derived at 18:09 (the time of totality on the previous day) and at 20:54 (a time close to the end 390 of observations on this day). Since the Ångström exponent for clouds is close to zero, the fact that the spectral dependence of the AODs (quantified with the Ångström exponent α) observed on 8 and 9 April is similar suggests that the AOD measurement on 8 April is fairly accurate and not greatly affected by clouds.





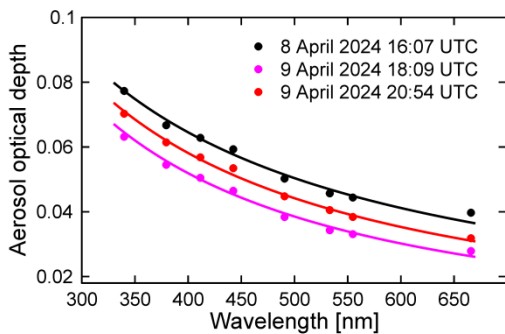

Figure 8: Aerosol optical depth measured in Mazatlán on 8 and 9 April 2024 (symbols). Lines are fit functions based on Ångström's turbidity formula using the following coefficients: 8 April 2024 16:07: α=1.1080, β=0.0234; 9 April 2024 18:09: α=1.3345, β=0.0153; 9 April 2024 20:45: α=1.2288, β=0.0189.

### 5.2.3    Total column ozone

Panels a–c of Figure 9 show TCO calculated from GUV, Microtops, and OMI measurements at Mazatlán for 7–9 April 2024. Data of 8 April were either processed "as is" or first corrected for the change in the extraterrestrial solar spectrum during the eclipse. Figure 9d shows the power spectrum density (PSD) of $TCO_{340/305}$ between 18:14:00 and 19:22:16 on 7 and 8 April 2024. The period starts shortly after the time of the 3rd contact and is 4096 s long. During a period of this length, fluctuations in TCO were observed by Mims and Mims (1993) and Zerefos et al. (2007) as discussed in Sect. 1.1. For comparison, Figure 9d also shows the PSD for a hypothetical scenario where TCO oscillates with an amplitude of 10 DU and a period of 60 s about a constant value of 300 DU. The following can be concluded from Figure 9:

- Uncorrected $TCO_{340/305}$ measurements increase by about 30 DU before totality and decrease by about the same amount thereafter. When the wavelength-dependent change in the extraterrestrial solar spectrum is taken into consideration, there is virtually no effect of the eclipse on TCO. Hence, the increase in TCO can be entirely explained with the LD effect.

- TCO values derived from GUV measurements on 7 and 8 April are affected by clouds, resulting in spurious variability. $TCO_{340/313}$ is much more impacted than $TCO_{340/305}$ because changes in ozone affect measurements at 305 nm much more strongly than at 313 nm. Changes introduced by ozone are therefore more important at 305 nm than 313 nm compared to cloud effects. Measurements on the clear-sky day 9 April 2024 are very consistent and vary smoothly with time, giving further confidence in the good quality of the GUV TCO data.

- The cirrus cloud cover on 7 and 8 April was similar and so are the fluctuations of TCO with time. On 8 April 2024, there is no obvious change in TCO that could be attributed to bow waves. However, because of the variability introduced by clouds, a small effect from the eclipse cannot be excluded but it would have to be smaller than about ± 2 DU or ± 0.7 %.



- During overcast conditions (2nd half of 8 April 2024), $TCO_{304/313}$ underestimate the actual TCO.

- There is a small increase in TCO over the course of the eclipse, which is both captured by LD-corrected GUV and Microtops measurements; however, this increase is comparable in magnitude to increases and decreases observed during other times. In fact, variations on 7 April tend to be larger than on eclipse day, 8 April.

- Scaled Microtops measurements are consistent with GUV measurements and do not indicate an increase at the time of totality or fluctuations beyond some scatter.

- On 9 April 2024, GUV measurements are higher by 2.4 % than the OMI measurements of this day. This is the typical difference between SUV-100 and OMI measurements observed at Antarctic sites equipped with SUV-100 spectroradiometers. The reason for this small bias is unknown.

- The PSDs for 7 and 8 April are virtually identical. There is no evidence of a periodic oscillation on 8 April that could have been triggered by bow waves. The PSD of the synthetic TCO (oscillation with an amplitude of 10 DU
and a periodicity of 60 s) suggests that oscillations with an amplitude of well below 10 DU should be detectable in a PSD calculated from measured TCO data, if such fluctuations exist.



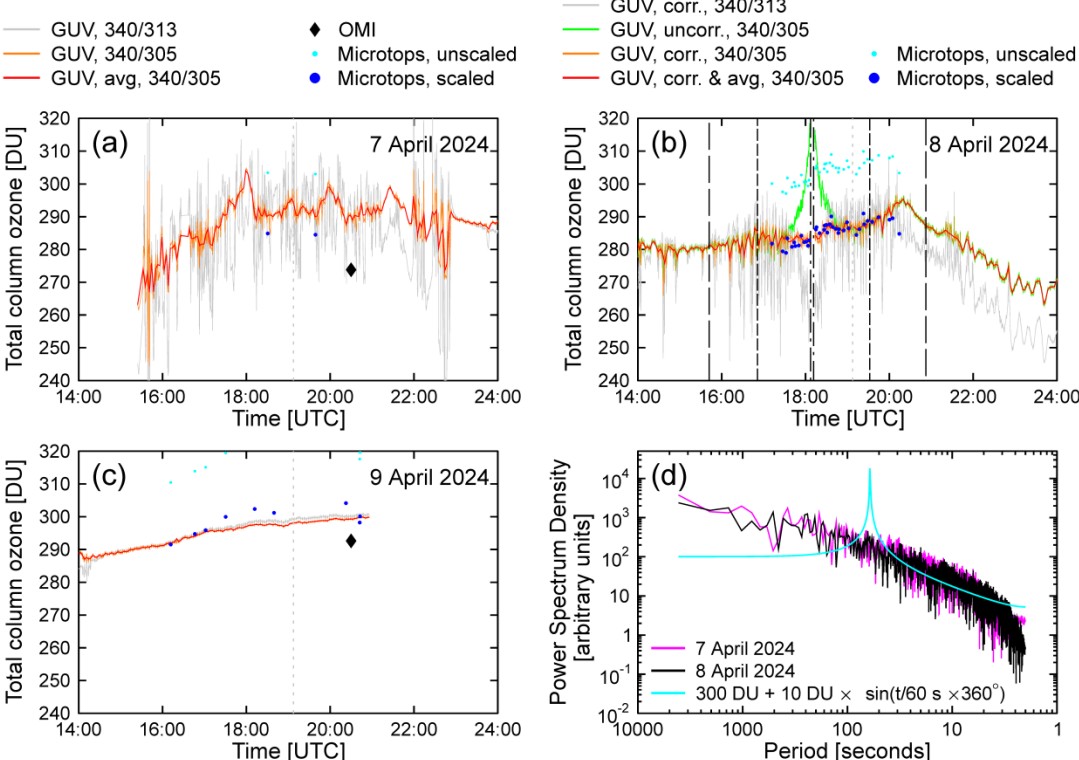

Figure 9: Measurements of TCO at Mazatlán on (a) 7 April 2024, (b) 8 April 2024, (c) and 9 April 2024. $TCO_{340/313}$ and $TCO_{340/305}$ measurements at 1 Hz are shown in gray and orange, respectively. The red line shows 90 second moving averages of TCO derived from $TCO_{340/305}$. Microtops measurements are biased high (cyan circles). Data scaled down by 6.5 % are also shown (blue) for ease of comparison with GUV data. OMI measurements are only available on 7 and 9 April 2024. All data were filtered for SZAs smaller than 75°. The vertical "very long-dashed" vertical lines (before 16:00 and at about 21:00) indicate the times when the shadow of the Moon either first hit the Earth (over the Pacific) or leaves the Earth (over Greenland). 1st and 4th contact at Mazatlán are indicated by dashed lines, and the 2nd and 3rd contact are indicated by dashed-dotted lines. Panel (d) shows the power spectrum density of $TCO_{340/305}$ between 18:14:00 and 19:22:16 on 7 and 8 April 2024. The PSD for a hypothetical scenario where TCO oscillates with an amplitude of 10 DU and a period of 60 s about a constant value of 300 DU is also shown (cyan line).

### 5.2.4 Variations in surface pressure

Figure 10a shows the surface pressure at Mazatlán on 7–9 April 2024 measured by the microbarometer. The pressure exhibits a similar diurnal cycle on the three days with an amplitude of about 1 hPa and a maximum at local solar noon. The diurnal cycle is caused by atmospheric tides resulting from the Sun's heating of the atmosphere during the day (Haurwitz and Cowley, 1973). Overlaid on this diurnal cycle are short-term fluctuation with an amplitude of about 0.2 hPa, which are also similar on the three days. There is no clear evidence that these variations are systematically different during the period




of the eclipse. Furthermore, the PDS of surface pressure for the three days (Figure 10b), which is based on a ~ 1 hour period starting at the time of maximum eclipse on 8 April, is also similar for the three days. Hence, there is no evidence that bow waves, which may have been excited by the eclipse, led to measurable changes in surface pressure.

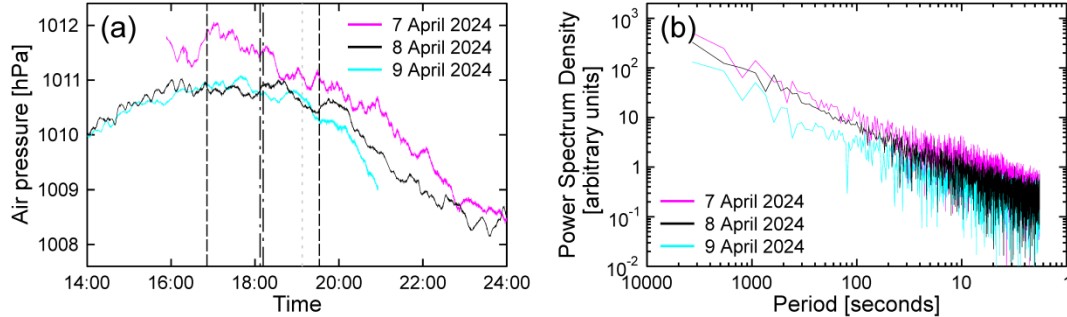

Figure 10: (a) Surface pressure at Mazatlán on 7–9 April measured by the microbarometer. 1st and 4th contact are indicated by dashed lines and the 2nd and 3rd contact are indicated by dashed-dotted lines. (b) Power spectrum density of surface pressure for the three days based on data collected between 18:09:25 (time to maximum eclipse on 8 April) and 19:11:02 on each day.

## 5.3 Measurements at Fort Collins

### 5.3.1 Spectral irradiance

Global spectral irradiance measured by the GUV radiometer in Fort Collins on 14 October 2023 are shown in Figure 11. The figure also presents measurements that were corrected for the change of the extraterrestrial solar spectrum by dividing the instrument's raw data by the change in the extraterrestrial irradiance before processing with GUVDP. The correction removes the effect of the eclipse with high fidelity, leading to a smooth curve in irradiance as one would expect on a day without an eclipse. Note that measurements become affected by cloud for times after 17:15. There is almost no indication of clouds between the start of the eclipse and this time.



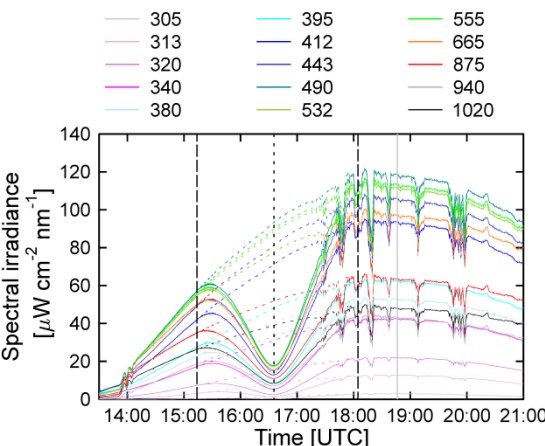

Figure 11: Global spectral irradiance measured on 14 October 2024 with the GUV radiometer (solid lines) and measurements corrected for the solar LD effect (broken lines). Vertical long-dashed lines indicate the 1st and 4th contact. The time of the eclipse maximum is indicated by a short-dashed line and the local solar noon is indicated by a gray line.

### 5.3.2    Aerosol optical depth

Figure 12a shows AOD at Fort Collins calculated from the direct irradiance derived with GUVDP using data acquired during shadow band sweeps. Without LD correction, AOD retrieved by the algorithm shows a large spurious peak centered at the time of maximum eclipse because the program attributes the reduced solar irradiance to aerosols. With LD correction, the effect of the eclipse is not noticeable in AOD data. Figure 12b shows AODs retrieved at three times during the eclipse (at 16:01, 16:35, and 17:01). Data are very consistent: AODs calculated at the time of the maximum eclipse (at 16:35) agree almost ideally (i.e., to better than 0.01) with AODs ~ 30 minutes before and after this time. This confirms again that the LD correction is accurate. Ångström's turbidity formula was fitted to the average of the three measurements, resulting in α=1.4917 and β=0.0284.



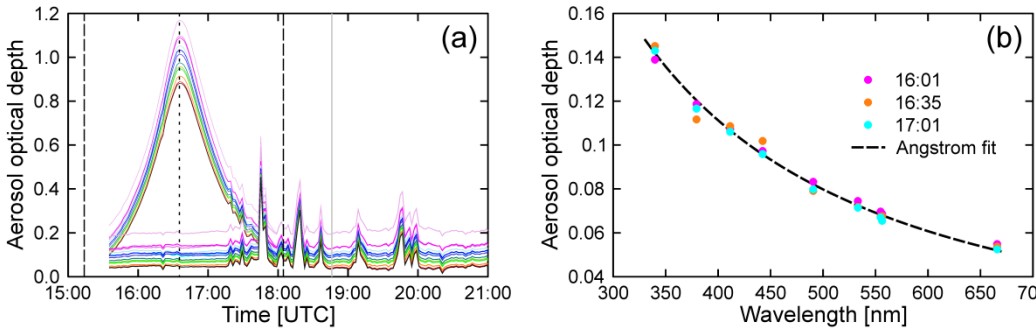

Figure 12: (a) Aerosol optical depth calculated from shadow band sweeps of the GUV radiometer at Fort Collins. Thin lines indicate spurious AODs measured without LD correction while thick lines indicate AOD corrected for the LD effect. The color coding is identical with that used in Figure 11. Data were not filtered for clouds, and spikes occurring after 17:15 are attributable to clouds. Long-dashed lines indicate the 1st and 4th contact. The time of the eclipse maximum is indicated by a short-dashed line and the local solar noon is indicated by a gray line. (b) Aerosol optical depth as a function of wavelength at three times during the eclipse. The broken black line shows the Ångström fit to the average of all data.

### 5.3.3     Total column ozone

Figure 13 shows TCO calculated from the GUV data on 14 October 2023. As was the case for Mazatlán, data were either processed "as is" or corrected for the LD effect. The following can be concluded:

- Uncorrected $TCO_{340/305}$ data increase by about 8 DU before totality and decrease by a similar amount thereafter. When data are corrected for the LD effect, $TCO_{340/305}$ remains constant to within $\pm$ 1 DU (0.4 %). Uncorrected $TCO_{340/313}$ data increase by 16 DU and LD-corrected data decrease by ~ 5 DU compared to the values at the start of the eclipse. (A decrease of similar magnitude is also hinted in the LD corrected data for Mazatlán (Figure 9b), although scatter from cloud effects make an assessment difficult.) The most likely reason for the overcorrection is the uncertainty of the LD coefficients near 313 nm published by Pierce and Slaughter (1977). The shape of the corrected dataset looks like an inverted LD function, strongly suggesting that the feature is caused by a systematic error in the LD correction instead of actual changes in TCO.

- TCO measurements derived after 17:15 are affected by clouds. As was the case for Mazatlán, $TCO_{340/313}$ is more affected than $TCO_{340/305}$.

- Early in the day (before 15:00), when SZAs are larger than 71°, $TCO_{340/313}$ is biased low.

- There are similar wavy patterns in corrected $TCO_{340/305}$ data close to the times of the 1st contact and maximum eclipse. The decrease in TCO is about 1 DU for the former and 1.5 DU for the latter (amplitude of 0.75 DU or 0.3 %). While these features could have been triggered by the eclipse, their magnitude is within the range of natural variability and it would be challenging to unambiguously attribute those to the solar eclipse.

- There are no fluctuations occurring on time scales of minutes.



- The day's OMI measurement is 4 DU (1.5 %) higher than GUV measurements.
- The data density is higher up to 15:35 because the shadow band was not operational before this time.

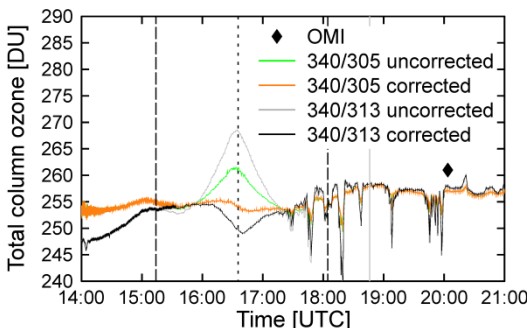

Figure 13: Measurements of TCO at Fort Collins on 14 October 2024 with the GUV. Uncorrected and LD-corrected $TCO_{340/305}$ data are shown in green and orange, respectively. Similar data for $TCO_{340/313}$ are shown in grey and black. The OMI measurement of this day is also indicated. Long-dashed lines indicate the 1[st] and 4[th] contact. The time of the eclipse maximum is indicated by a short-dashed line and the local solar noon is indicated by a gray line.

## 5.4 Union Glacier

### 5.4.1 Spectral irradiance

Figure 14a shows cosine error corrected measurements of global spectral irradiance by the GUV-3511 radiometer at Union Glacier performed on 4 and 5 December 2021, plus model calculations with input parameters optimized for 4 December.
Specifically, TCO was set to 215 DU (the average TCO on this day measured by OMI) and AOD was parameterized with Ångström's turbidity formula by setting the Ångström coefficients α and β to α=1.26 and β=0.0061. These parameters were determined by fits to AOD data measured with the AERONET sunphotometer (Sect. 5.4.2). For wavelengths up to 670 nm, albedo was set to 0.95 in accordance with measurements by Cordero et al. (2014). Albedo at 875, 1,020, 1,245, and 1,640 nm was set to 0.88, 0.75, 0.58, and 0.2 based on measurements at the South Pole (Grenfell et al., 1994). Figure 14b plots the
ratio of measurements on 4 December (with and without the LD correction applied) and the model. Lastly, Figure 14c presents the ratio of measurements on 4 December and 5 December, again with and without the LD-correction applied. As the SZAs were different on the two days, measurements on 5 December 2021 were interpolated in SZA to match the SZA on 4 December 2021 before calculating the ratios.





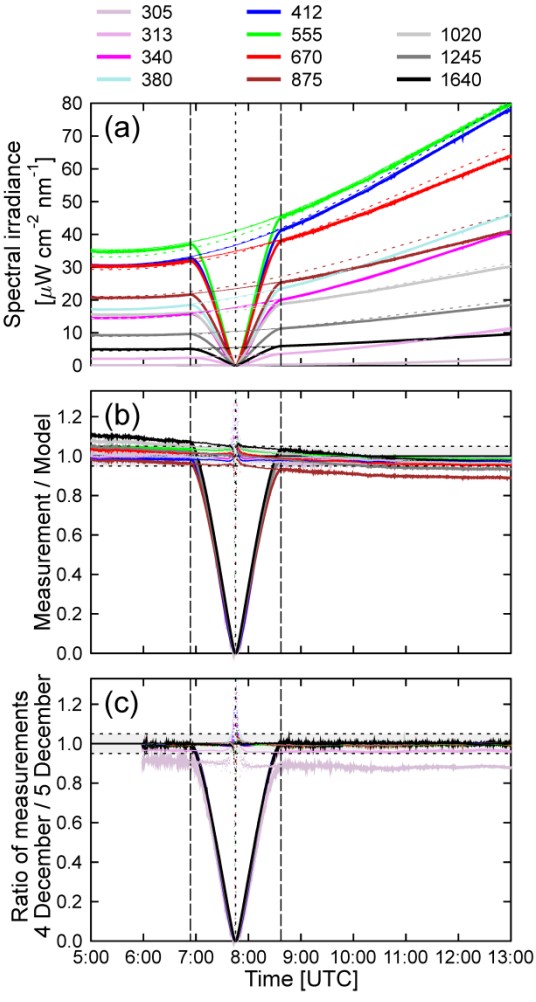

Figure 14: (a) Global spectral irradiance measured at Union Glacier on eclipse day 4 December 2021 (thick lines) and on 5 December 2021 (thin lines), and the modeled spectral irradiance with input parameters optimized for 4 December 2021 (broken lines). (b) Ratio of measurement and model for 4 December 2021 (solid lines), plus ratio of LD-corrected measurement and the model (dots). (c) Ratio of measurement on 4 December 2021 and 5 December 2021 (solid lines), plus ratio of LD-corrected measurement on 4 December 2021 and measurement on 5 December 2021 (dots). Long-dashed lines in all panels indicate the $1^{st}$ and $4^{th}$ contact. The time of the eclipse maximum is indicated by a short-dashed line. The horizontal dotted black lines in panels (b) and (c) indicate the region of ± 5 % about the ideal ratio of one.

Figure 14 allows the following conclusions:

- Both 4 and 5 December 2021 were cloudless days with stable atmospheric conditions. This is particularly obvious from the ratio of measurements of the two days shown in Figure 14c. Before and after the eclipse, measurements agree to within ± 1 % for wavelength between 340 and 1,640 nm. Observations at 313 nm and 305 nm were lower on 4 December by about 4 % and 10 %, respectively, because of the larger TCO on 4 December (OMI measured 215 DU on 4 December and 198 DU on 5 December).

- Before the $1^{st}$ contact and after the $4^{th}$ contact, measurements at wavelength between 305 and 412 nm agree with the model to within ± 4 % (Figure 14b). At longer wavelengths, ratios between measurement and model decrease with time but stay within ± 11 % of unity. This bias and downward trend are due to a systematic error in the cosine error correction. As mentioned in Sect. 3.1, the angular response functions of the GUV used at Union Glacier were not



characterized and generic response function were used for the correction instead. At a SZA of 75°, the effect of cosine errors are typically the largest and the difference between the actual (but unknown) and generic response functions can explain the pattern apparent in Figure 14b. However, this has no consequence on assessing the effect of the eclipse on changes in TCO as the affected wavelengths are not used for the ozone retrieval.

- At wavelengths in the UV range, the LD-corrected measurements agree with model calculations to within ± 5 % up
to 5 minutes before the 2nd and after the 3rd contact. This period decreases to 1.5 minutes for wavelengths in the visible and infrared range, confirming again the accuracy of the LD-correction method. At times close to totality, measurements exceed results of the one-dimensional (1-D) model. Assessing these differences would require a 3-D model, which is beyond the scope of this paper.

Figure 15 shows a close up of the spectral irradiance within ± 5 minutes of totality. A comparison with the similar plot for Mazatlán (Figure 7) reveals that the change in spectral irradiances at 305 nm and 313 nm are much smaller at Union Glacier than at Mazatlán. This difference is likely caused by the fact that the eclipse at Union Glacier occurred at a high-albedo site and at a large SZA of 76°. Both factors lead to almost entirely diffuse radiation at these wavelengths (the direct contribution is less than 4 % at these wavelengths according to our model calculations). In contrast, spectral irradiance at
1,245 and 1,640 nm exhibit a steep decrease when transitioning into totality with measurements dropping below the detection limit.

Figure 15: Close up of change in spectral irradiance at Union Glacier within ±5 minutes of totality. The color coding is identical with that used in Figure 14. The times of the 2nd and 3rd contact are indicated by dashed-dotted lines and the time of maximum eclipse is indicated by a short-dashed line.

### 5.4.2 Aerosol optical depth

Figure 16 shows AODs at Union Glacier measured by the AERONET sunphotometer (Sect. 3.3). As it was the case for Fort
Collins, AODs at Union Glacier calculated without LD correction show a large spurious peak (Figure 16a) while LD-corrected data change by less than ± 0.01 before totality and by less than ± 0.02 after totality (Figure 16b). These small




variations could be caused by the low signal levels during the eclipse or by the fact that the unconcluded portion of the Sun during the eclipse is not in the center of the instrument's field of view. AODs shortly before the 1st contact at 6:47, shortly after the 4th contact at 8:37, and one hour later are consistent to within 0.004 and smaller than 0.025 at 340 nm (Figure 16c).

An Ångström fit to the data results in α=1.26 and β=0.0061. Measured AODs show more variability about the fit line compared to variations observed in Fort Collins. This can be explained by the fact that AODs at Union Glacier are very small and comparable to the uncertainty of AERONET data of 0.01–0.021 (Sinyuk et al., 2020; Eck et al., 1999). These data confirm that the eclipse at Union Glacier occurred under pristine conditions with AODs close to Antarctic background conditions (Chen et al., 2024).


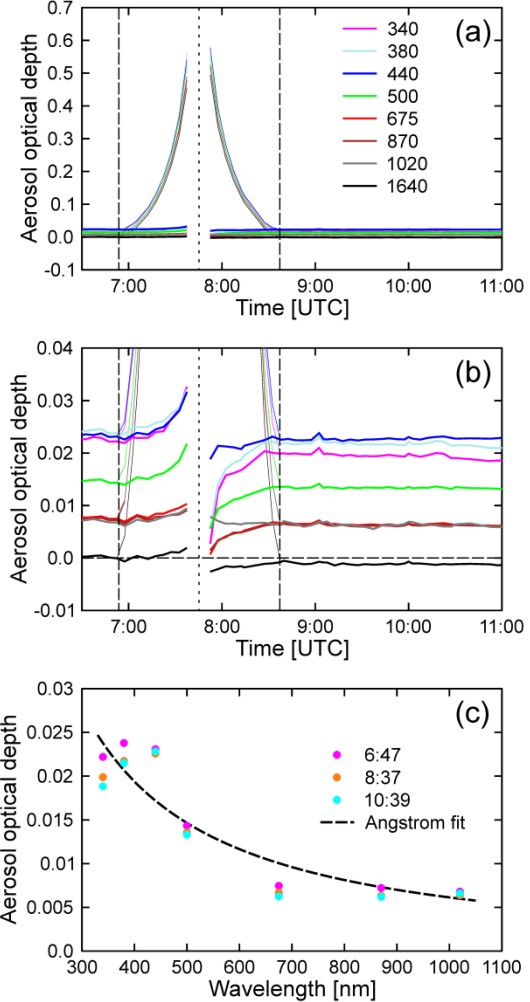

Figure 16: (a) Aerosol optical depth measured by AERONET sunphotometer at Union Glacier. Thin lines indicate spurious AODs measured without LD correction while thick lines indicate AOD corrected for the LD effect. (b) Same as (a) but plotted on a smaller y-scale to emphasize AODs after LD correction. Vertical long-dashed lines in Panel (a) and (b) indicate the 1st and 4th contact. The time of the eclipse maximum is indicated by a short-dashed line. (c) Aerosol optical depth as a function of wavelength for three times. The broken black line shows the Ångström fit to the average of all data.



### 5.4.3 Total column ozone

Figure 17 shows TCO calculated from the GUV data collected at Union Glacier on 4 December 2021. As before, data were either processed "as is" or corrected for the LD effect. The following can be concluded:

- TCO$_{340/305}$ data are much noisier than TCO$_{340/313}$ data because of the large SZAs at Union Glacier (e.g., SZA=77.2° at the time of the 1$^{st}$ contact). At these large SZAs, the spectral irradiance at 305 nm is less than one order of magnitude above the instrument's detection limit. While TCO$_{340/305}$ data are typically more accurate than TCO$_{340/313}$ data, because they are less affected by factors other than ozone, use of TCO$_{340/313}$ becomes advantageous if the SZA is very large.

- The LD effects leads to a spurious increase in TCO when approaching totality as it was observed at Mazatlán and Fort Collins. However, within ± 3.5 minutes of totality, the calculated TCO drops greatly. The drop can be seen both in TCO$_{340/305}$ and TCO$_{340/313}$ but it is much stronger for the latter.

- TCO$_{340/305}$ data corrected for the LD effect scatter to within ± 2.5 DU outside of a ± 7 minutes interval centered at the maximum eclipse. LD-corrected TCO$_{340/313}$ data are more constant than uncorrected data outside of about ± 15 minutes from the maximum eclipse, but the correction overcorrects closer in time to totality.

- Apart from these changes in TCO close to totality, there is no evidence of oscillations in TCO that could be attributed to gravity waves. Closer inspection of TCO$_{340/313}$ data during a period of four hours starting with totality establishes an upper limit for short-term fluctuation in TCO of ± 0.05 DU (± 0.03 %), which is well within the range of natural variability.





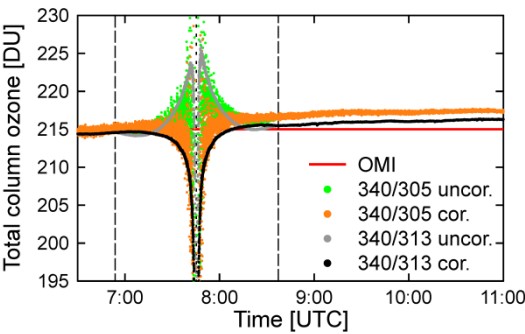

Figure 17: GUV measurements of TCO at Union Glacier on 4 December 2021. Uncorrected and LD-corrected TCO$_{340/305}$ data are shown in green and orange, respectively. Similar data calculated for TCO$_{340/313}$ are shown in grey and black. The average of OMI measurement for this day is indicated by a horizontal red line. Long-dashed lines indicate the 1$^{st}$ and 4$^{th}$ contact. The 3$^{rd}$ and 4$^{th}$ contact are indicated by dashed-dotted lines.

## 5.5 Reduction in spectral irradiance during totality

Figure 18 quantifies the decrease of spectral irradiance during totality by plotting "reduction factors", defined as the ratio of spectral irradiance expected without the eclipse and the measurement at the time of totality, for three total solar eclipses: the ones at Mazatlán and Union Glacier discussed here and the one observed at Smith Rock State Park in Oregon on 21 August 2017 described by B&P19. Reduction factors calculated for Mazatlán are generally larger than those for Oregon. The effect is particularly strong at wavelengths where ozone absorption is large (313 and 320 nm, with 305 nm in the noise) or at 940 nm, which is affected by water vapor absorption. This can be explained by the longer path lengths for photons entering the atmosphere outside the larger shadow on 8 April 2024. (The duration of the 8 April 2024 eclipse was 4:21 minutes while that of the 21 August 2017 eclipse was only 2:35 minutes. According to JPL Horizon data, the ratio of the diameters of Moon and Sun during totality were 1.0574 and 1.0250 on the two days, respectively. The area of the Moon's shadow was therefore larger on 8 April 2024, which would imply that it was darker during totality in 2024 compared 2017.)

Reduction factors for Union Glacier are considerably smaller compared to the those at the other two sites. This may have several reasons, such as the short duration of the eclipse (48 seconds), the large SZA (76.1°), and the high surface albedo (up to 0.95 in the UV range). Interestingly, the reduction factor at 305 nm is lower than those of visible wavelengths while the opposite is true for reduction factors at Mazatlán and Oregon. This is likely a consequence of the large SZA leading to almost entirely diffuse radiation at 305 nm at the surface. We speculate that most photons detected at the ground and at this SZA have entered the atmosphere far away from the observation site and were first scattered above the ozone layer. If that were the case, the shadow of the Moon would affect only a relative small part of the area where photons of this wavelength enter the atmosphere and subsequently travel towards the surface. Quantifying this potential effect would require 3-D radiative transfer calculations such as those performed by Ockenfuß et al. (2020).





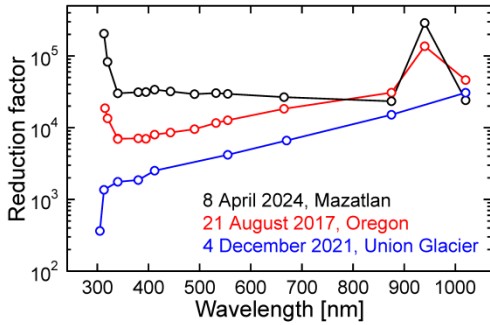

Figure 18: Reduction factors defined as the ratio of spectral irradiance expected without the eclipse and the measurement at the time of totality. The red curve shows the factor for 21 August 2017 and is identical to that shown in Figure 15 of B&P19. The black curve is the factor determined for the eclipse in Mazatlán on 8 April 2024. The spectral irradiance at the time of totality used for this plot is the average of measurements within ± 30 seconds of the time of the eclipse maximum. Measurements at 305 nm are below the detection limit at Mazatlán and Oregon and are not shown. Similarly, measurements at 1,245 and 1,640 nm at Union Glacier are in the noise.

## 6 Discussion

Some changes in TCO during solar eclipses reported in the past are in conflict with results presented above. The potential reasons for these discrepancies are discussed below.

We found no convincing evidence of short-term fluctuations in TCO for the three eclipses discussed here, consistent with
results of our earlier publication describing the 2017 total solar eclipse (B&P19). The upper limit of the amplitude of oscillations in TCO observed at Union Glacier and Fort Collins were 0.03 % and 0.3 %, respectively. The limit for Mazatlán is somewhat larger due to compounding effects of clouds. These variations are well within the natural variability and the uncertainty of our measurements. We also found no evidence of short-term fluctuation in surface pressure that could be attributed to the effect of gravity waves. While it cannot be excluded that conditions during past eclipses where large
fluctuations in TCO were observed (e.g., Bojkov, 1968; Mims and Mims, 1993; Zerefos et al., 2007; Zerefos et al., 2000) were different, we find that this is unlikely. For example, we found no fluctuations in TCO during the 2017 eclipse (B&P19) even though bow waves were clearly apparent in the ionosphere at this time (Zhang et al., 2017). Furthermore, totality at Mazatlán occurred at a small SZA of 20.9° and lasted quite long (4 minutes and 21 seconds). These conditions should have been ideal for the development of gravity waves because of the large absolute change in the solar flux and the resulting
cooling of the upper and lower atmosphere over a large area. At Union Glacier, totality was relatively short (48 seconds) but the eclipse occurred only 17 days before the austral summer solstice, leading to significant changes in air temperature across the continent. For example, the 2 m air temperature dropped by about 3 °C at Union Glacier and by almost 5 °C at the South Pole (Garreaud et al., 2023). Yet, no significant changes in the TCO were detected at both Mazatlán and Union Glacier.





Increases in $TCO_{340/305}$ within a few minutes of the eclipse maximum can be convincingly explained with the LD effect at
Mazatlán and Fort Collins. The situation at Union Glacier is more complicated because of low solar elevations and high
albedo. Yet even at this location, LD-corrected $TCO_{340/305}$ data are quite constant up to five minutes before and after the
period of totality. Measurements closer to totality are affected by noise in the measurements at 305 nm and three-
dimensional effects (see below). While the LD effect has been acknowledged by several authors in the past, its magnitude is
typically regarded as too small (i.e., < 0.01 % (Blumthaler et al., 2006; Kazadzis et al., 2007), < 1 % (Zerefos et al., 2000), or
< 1.6 % (Kazadzis et al., 2007)) to quantitatively explain the sometimes large increases in TCO reported in these
publications. In most of these cases, the LD effect is underestimated because of the use of the LD parameterization by
Waldmeier (1941), which has been shown to be too small (B&P19 and Figure 1).

Increases in $TCO_{340/313}$ at the time of totality are larger than increases in $TCO_{340/305}$, and the LD correction overcorrects
$TCO_{340/313}$ data at the three sites. The likely reason for this overcorrections are systematic errors in the coefficients published
by Pierce and Slaughter (1977), which are the basis of the LD correction. This assertion is supported by the observation that
the spectral shape of the uncorrected and corrected data is very similar, albeit inverted. It is highly unlikely that effects that
could lead to a real change in TCO over a short periods of minutes before and after totality would have the same spectral
dependence as the LD effect, which is independent of processes in the Earth's atmosphere as it is entirely determined by the
temperature structure of the Sun's photosphere. Furthermore, there is no plausible physical process that could increase the
TCO by several tens of DU within a few minutes and then cause a symmetrical decrease by the same amount after totality, in
particular when considering that the day/night change in TCO is less than 0.6 % (Sect. 1.4).

Measurements of TCO by the Microtops are about 6.5 % too large, both at San Diego (Figure 5) and Mazatlán (Figure 9).
We attribute this bias to the calibration of the instrument even though it was acquired just before the campaign. However,
changes in TCO over time measured by the Microtops and GUV at the two sites are very consistent. Furthermore, changes in
Microtops TCO data (scaled by 1/1.065) over time agree almost ideally with LD-corrected $GUV_{340/305}$ data at Mazatlán.
However, it is surprising that Microtops data are not affected by the LD effect (Figure 9b). This may partly be due to the fact
that the Microtops uses a smaller wavelength range (305–320 nm) than the GUV (305–340 nm) over which the wavelength-
dependence of the LD effect is smaller. Since the instrument calculates the TCO separately from the 305/312 nm pair and the
312/320 nm pair and then combines the two results, it is conceivable that LD effects of the two bands partly cancel when
merging their results. Whatever the reasons, it is reassuring that Microtops measurements do not indicate variations in TCO
that could be attributed to the eclipse, consistent with the GUV results.

Fluctuations in surface pressure during an eclipse can only be observed during stable atmospheric conditions (Sect. 1.3).
This was the case at Mazatlán because the diurnal cycle in pressure observed on the three days were almost identical both in
magnitude and duration (Figure 10a). Yet, data don't reveal any evidence of oscillations that could be attributed to gravity
waves.



Uncorrected $TCO_{340/313}$ data at Union Glacier increase by about 8 DU (4 %) leading up to totality because of the LD effect and drop sharply by about 40 DU within 2.5 minutes before the $2^{nd}$ contact. Three-dimensional (3-D) Monte Carlo model calculations by Emde and Mayer (2007) simulating the total solar eclipse of 29 March 2006 showed that 1-D radiative transfer calculations, which also formed the basis for the ozone lookup tables used here (Sect. 4.1), are no longer reliable
about 1.5 minutes before the $2^{nd}$ contact. (The period applicable to the eclipse at Union Glacier could be longer). Furthermore, by modeling the radiation of the 2017 total solar eclipse with a 3-D model similar to that used by Emde and Mayer (2007), Ockenfuß et al. (2020) found that the spectral UV irradiance close to totality becomes very sensitive to the vertical distribution of ozone in the atmosphere, surface albedo, and topography. These factors likely played a role also here. Hence, the drop in $TCO_{340/313}$ is an artifact of using a lookup table that is inappropriate close in time to totality. We speculate
that the sharp decrease in $TCO_{340/313}$ is caused by photons that first traversed through the upper atmosphere horizontally (from a region not shaded by the Moon towards a point approximately above the site of observation), and then were scattered downward and passed through the ozone layer more vertically. This would lead to a shorter optical path length through the ozone layer compared to uneclipsed conditions. Resolving this issue quantitatively would require 3-D calculations, which are beyond the scope of this paper. It is clear, however, that the sharp apparent decrease in TCO and the immediate rebound after
totality is far too fast and large in magnitude to be real.

Apart from studying variations in TCO, we also quantified the spectral irradiance during totality at Mazatlán and Union Glacier and calculated the AOD at three sites. At Mazatlán, spectral irradiance in the visible range is reduced by a factor of 30,500 on average during totality, which is a larger reduction that that observed during the 2017 total solar eclipse (B&P19). This can be explained with the larger eclipse magnitude of the 2024 eclipse, which led to a larger shadow of the Moon in
2024. Conversely, the reduction at Union Glacier was considerably lower than at the other two sites. At Fort Collins and Union Glacier, the uncorrected AOD exhibits a large peak centered at the time of eclipse maximum. After correcting for the LD effect, the AOD remains almost constant over the period of the eclipse, providing further evidence of the fidelity of the LD-correction developed by B&P19.

## 7  Conclusions

Measurements of spectral irradiance at 1 Hz were performed with GUVis-3511 multi-filter radiometers during solar eclipses observed at Mazatlán, Fort Collins, and Union Glacier. GUV measurements at Mazatlán were augmented by observations with a Microtops ozonometer and a microbarometer. At all sites, TCO retrieved from the data peaked at the time of maximum eclipse, but these increases could be convincingly explained by solar limb darkening, which describes the wavelength-dependent decrease in the Sun's brightness between the Sun's center and its edge. We found no evidence of
periodic oscillations or other short-term fluctuations in TCO or surface pressure within the hours before or after the eclipses that exceed natural variability. The upper limit of the amplitude of oscillations in TCO observed at Union Glacier, Fort Collins, and Mazatlán were 0.03 %, 0.3 % and 0.7 %. The relative high value at Mazatlán is due to variability introduced by



changing clouds. Our observations contradict reports of much larger fluctuations during previous eclipses published in the literature. We conclude that these large variations were either caused by an insufficient correction for the solar limb darkening effect, measurement artifacts, data processing issues, or a combination thereof. This conclusion is also supported by the fact that day/night differences in TCO are smaller than 2 DU (~ 0.6 %) at mid-latitudes and there is no known physical mechanism that could greatly amply this magnitude during the relative short period of a solar eclipse. Assessing variations in TCO within ~ 2 minutes of the start or the end of totality would require 3-D model calculations, which are beyond the scope of this paper. We therefore did not assess potential change in TCO within in this short period. However, 3-D radiative transfer simulation of spectral solar irradiance complementing the solar eclipse at Mazatlán are currently underway and these new results may provide further insights in changes in TCO close to totality.

We answer the question "Does total column ozone change during a solar eclipse?" posed in the title with "only by a very small amount and perhaps not at all", since variations in the order of ± 0.3 % cannot be excluded. However, the cross-correlation between variations in TEC and JO[1]D described by Zerefos et al. (2007) and summarized in Sect. 1.1 remains unexplained. While this correlation is robust, the reported amplitudes of 5–10 DU in TCO and 3–5 % in JO[1]D are well outside the range of our measurements.

**Code and data availability**

GUVis-3511 Data Processor software and manual, raw data, and data shown in Figure 4 through Figure 18 are available at https://www.dropbox.com/scl/fo/xhgtb7gydinjdaplezfr1/AGpGzj2W7yYAl_pzlqmSQGs?rlkey=f9yatzl7m2qyn5tojvmxv3l08&st=da7ooulz&dl=0. Upon acceptance of the manuscript, all assets will be transferred to Zenodo (https://about.zenodo.org/) and a Digital Object Identifier (DOI) will be assigned to the dataset.

**Author contributions**

GHB devised the study, executed the measurements at Mazatlán, performed all data analyses, and wrote the manuscript. GTJ and SS performed the measurements at Fort Collins. RRC oversaw that measurement at Union Glacier. EISA, JJ, and JAR executed the measurements at Union Glacier. EISA also calibrated the GUV in 2015. RNL oversaw the project.

**Competing interests**

GHB is employed by Biospherical Instruments, Inc., which is also the manufacturer of the GUVis-3511 radiometer described in this paper.

**Acknowledgements**

This research was funded by the Division of Atmospheric and Geospace Sciences (AGS) of the U.S. National Science Foundation (NSF), grant number 2328210. Any opinions, findings, and conclusions or recommendations expressed in this manuscript are those of the authors and do not necessarily reflect the views of the NSF. We thank Ms. Anne L. Hoppe for



coordinating travel to Mazatlán, transportation services, help setting up the instruments, and performing measurements; Dr. Anastazia T. Banaszak from the Institute of Marine Sciences and Limnology of the National Autonomous University of Mexico at Puerto Morelos, Mexico, for facilitating the project; Dr. Felipe Amezcua Martínez and his staff of the Institute of Marine Sciences and Limnology of the National Autonomous University of Mexico at Mazatlán, Mexico, for his permission to set up the instruments at his institute, his hospitality, and help with logistics; Paul Maley, expedition coordinator for the NASA Johnson Space Center Astronomical Society, Houston, Texas, for sharing his knowledge on observing solar eclipses in general and aiding the observations at Mazatlán; and Ms. Graciela Alvarez and Ms. Rosa Vazquez from the U.S. embassy in Mexico City for facilitating the permitting process with the Mexican government. We are further grateful to Mr. Luis Gerardo Esparza Ríos, General Director of the Department of Geography and Environment at the National Institute of Statistics and Geography (INEGI), Mexico; José Arturo Sánchez Monterrubio, Deputy Director of aerial surveys and geographical explorations of the General Directorate of Geography and Environment of Mexico; and José Alfredo Galvan Corona, General Director of Project Operations in Mexico for granting permission to conduct scientific research in Mexico and to perform measurements at Mazatlán. RRC, EISA, JJ, and JAR acknowledge the support of the Agencia Nacional de Investigacion y Desarrollo (ANID) of Chile (grant number ANILLO ACT210046) and the Chilean Antarctic Institute (INACH), grant number Preis RT_69-20.

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
