# Peer review of "Does total column ozone change during a solar eclipse?"

_EGUsphere, 2024_

## Author Comment (AC1)

**Response to comments by Anonymous Referee #1**

We thank the referee for his or her comments, which we have addressed as follows:

**Comment by Referee**

The study by Bernhard et al discusses the variability in total ozone during three solar eclipses. The authors use an improved parameterization, relative to previous studies, for the Limp Darkening correction, which results in a more accurate total ozone retrieval (from direct sun ground based spectral measurements) during solar eclipses. They finally show that the variability in total ozone during such events is much smaller than what has been proposed in previous studies. The study contributes significantly to the understanding of processes that take place in the stratosphere during solar eclipses.

The paper is well written, well structured, and within the scope of the journal.

**Authors' Response**

Thank you for these kind remarks.

**Change to manuscript**

None.
* * *
**Comment by Referee**

Minor comments and suggestions for technical corrections:

In addition to the studies that have been discussed and cited by the authors there are a few more recent studies that could be discussed in the introduction. For example:

https://www.mdpi.com/2072-4292/16/1/14

https://www.sciencedirect.com/science/article/pii/S1309104221001823

**Authors' Response**

There are many papers that address changes in ozone and other atmospheric parameters that occur during a solar eclipse and it would be impossible to cite all relevant papers. We agree, however, with the referee that the two papers suggested by him or her would add valuable information to the paper as they address changes in the vertical distribution of ozone during a solar eclipse (which has not been discussed in our manuscript) and the importance of natural variability and dynamics, which often obscure variations caused by a solar eclipse. We will therefore add the paragraph indicated below to the manuscript.

**Change to manuscript**

The following text will be added at the end of Section 1.1:

> "Lastly, variations of ozone concentrations during the annular solar eclipse of 21 June 2020 have recently been reported in the upper stratosphere and mesosphere based on observations by the Microwave Limb Sounder (MLS) on NASA's Aura satellite (Li et al., 2023). Accordingly, ozone concentrations slightly decreased near 40 km between 24° N and 36° N near 90° E, and increased conspicuously above 45 km, particularly between 60 and 65 km. These changes are comparable in magnitude with the effect of the day/night cycle on ozone concentrations at these altitudes discussed in Sect. 1.4. However, since approximately 99 % of ozone is at altitudes below 45 km, the effect on TCO is expected to be small. We also note that the MLS dataset is not well suited for observing short-lasting phenomena such as a solar eclipse because the instrument provides only few measurements during the period and over the geographic region of interest. It is therefore difficult to separate dynamical and natural occurring effects from those caused by an eclipse. The challenge in distinguishing between dynamical and eclipse-related processes was also emphasized by Akhil Raj and Ratnam (2021) who discussed changes in the vertical distribution of ozone that they observed during the annular eclipse of 26 December 2019 over India."
* * *
**Comment by Referee**

L342: Do the authors mean "critically dependent"?

L500: Should it be "functions" instead of "function"?

**Authors' Response**

The two typos were corrected as suggested by the referee.

---

## Author Comment (AC2)

**Response to comments by Anonymous Referee #2**

We thank the referee for his or her comments, which we have addressed as follows:

**Comment by Referee**

Review of the paper: "Does total column ozone change during a solar eclipse?" by Germar H. Bernhard et al.

General comments

This manuscript studies the short-term variability of the total ozone column (TCO) during solar eclipses in order to find out if the variations in this magnitude are real or, by contrary, the observed variability is derived from instrumental errors. For this goal, the authors work with TCO measurements recorded during three solar eclipses by GUVis-3511 and Microtops instruments. The topic is highly interesting and appropriated for ACP journal. In my opinion, the manuscript is clear and well written. Nevertheless, the following specific comments must be addressed by the authors before its final publication

**Authors' Response**

Thank you for these kind remarks. We will address the specific comments as discussed below.

**Change to manuscript**

None.
* * *
**Comment by Referee**

1. Section 4.1. TCO values are derived by GUVis-3511 from the wavelength pairs of 340/305 and 340/313 nm. The authors should justify this selection, for example, giving some references in which comparations of the TCO estimations by GUV instruments using different wavelength pairs against reference data were reported (e.g. Piedehierro et al., 2017). It must be noted that Dahlback (1996) proposed 320/305 pair, being taken as reference for later studies and adopted in the NILU-UV product software. Why this pair is not used for the present study?. For instance, the authors could obtain TCO values using 320/305 pair and these values be compared against SUV-100 spectroradiometer (subsection 5.1), following the comparison reported for TCO values derived from the wavelength pairs of 340/305 and 340/313 nm.

**Authors' Response**

We derived TCO from the 340/305 pair because this is the pair originally used by Stamnes et al. (1991) who developed the method of deriving TCO from measurements of spectral global (Sun and sky) irradiance. In that study, Stamnes et al. (1991) also showed that the effects of clouds on the retrieved TCO values is small. Piedehierro et al. (2017) calculated TCO from different (320/305, 320/313, 340/305, 340/313) pairs for a GUV radiometer and a NILU-UV radiometer (an instrument that is very similar to a GUV) and concluded that TCO retrievals based on the 340/305 pair should be used because they agree best with reference TCO values derived from direct-Sun measurements of a collocated Brewer spectrophotometer. Indeed, having a large spread between wavelengths that absorb ozone weakly (e.g., 340 nm) and strongly (e.g., 305 nm) has the advantage that calibration uncertainties have a smaller effect on the retrieved TCO compared to retrievals that use wavelengths that are closer together. However, when the solar zenith angle is very large (as it was the case during the eclipse at Union Glacier), measurements at 305 nm are close to the detection limit and measurements at 305 nm are therefore replaced with a measurements at 313 nm in our work, consistent with the recommendation by Piedehierro et al. (2017) and the method used in the NILU-UV processing software.

Having said this, we note that the study by Piedehierro et al. (2017) is based on cloud-free days only and the study by Stamnes et al. (1991) does not consider wavelength pairs other than 340/305. It therefore cannot be inferred from these studies whether TCO retrievals using the 320/305 pair are less sensitive to clouds that those based on the 340/305 pair.

We agree with the referee that computer simulations performed by Dahlback (1996) showed that the least influence of clouds on the derived total ozone abundance is obtained with the 320/305 ratio, and we should have taken this conclusion into consideration when drafting the manuscript. As suggested by the referee, we have therefore recalculated TCO values at San Diego, Fort Collins, and Mazatlán with this pair. (The GUV used at Union Glacier did not have a channel at 320 nm; however, the location was cloud-free, so changing from the 340/305 pair to 320/305 should have made little difference if measurements at 320 nm existed). We conclude from our results that the effect of clouds is indeed reduced when using the 320/305 instead of the 340/305 pair. Furthermore, retrievals from both pairs during clear sky conditions are consistent, confirming that calibration uncertainties are small.

We have therefore added TCO retrievals based on the 320/305 pair to Figures 5 (San Diego), 9 (Mazatlán) , and 13 (Fort Collins). We also changed the color scheme of Figure 17 (Union Glacier) to be consistent with that of the other sites. As described in more detail below ("Change to manuscript"), we also added new text to describe the features of the updated plots.

Most importantly, based on the new results, we reduced the upper limit of the amplitude of oscillations in TCO observed at Mazatlán from 0.7 % to 0.4 %, which is only slightly larger than the threshold of 0.3 % determined for Fort Collins (which was cloud free).

In conclusion, by changing TCO retrievals from the 340/305 pair to the 320/305, the effect of clouds becomes indeed less and the evidence that the solar eclipse at Mazatlán did not lead to variations in TCO beyond natural variability becomes even stronger.

**Change to manuscript**

- The upper limit of the amplitude of oscillations in TCO observed at Mazatlán will be changed from 0.7 % to 0.4 %.
- The following will be added to the end of Section 4.1 (Calculation of total column ozone):

> "TCO data calculated from the 340/305, 340/313, and 320/305 pairs are referred to as $TCO_{340/305}$, $TCO_{340/313}$, and $TCO_{320/305}$, respectively. In general, $TCO_{340/305}$ data are the most accurate data of the three datasets for clear-sky conditions because they are least impacted by calibration uncertainties (Piedehierro et al., 2017). However, $TCO_{340/313}$ data become more accurate at high SZAs when measurements at 305 nm are close to the detection limit. Lastly, simulations by Dahlback (1996) showed that $TCO_{320/305}$ data are least influenced by clouds. Hence, this dataset should be the most suitable for Mazatlán, the site impacted by cirrus clouds."

- Figure 5 will be replaced with the following figure and the figure caption will adjusted accordingly:

[Figure]

Note that this figure now indicates TCO values derived from the 340/305 and 320/305 pairs in blue and orange, respectively.

To reflect these changes, the text describing the figure will be changed to:

> "For the clear-sky period, TCO data derived from the GUV for SZA < 80° are biased low relative to the SUV-100 data by −1.1 %, −1.1 %, and −0.9 % for $TCO_{340/305}$, $TCO_{340/313}$, and $TCO_{320/305}$, respectively. The corresponding relative standard deviations are 0.6 %, 1.0 %, and 0.9 %, respectively. For the cloudy period, the standard deviations are increased to 2.4 %, 3.5 %, and 2.2 %, respectively. […].$TCO_{320/305}$ is least affected by clouds, confirming the conclusion by Dahlback (1996) mentioned earlier."

- Figure 9 will be replaced with the following figure and the figure caption will adjusted accordingly:

[Figure]

To reflect these changes, it will be noted in the text that $TCO_{320/305}$ is least impacted by cloud effects, and as a consequence, the upper limit of fluctuations in TCO due to the eclipse is reduced to ± 1.2 DU or ± 0.4 % (changed from ± 2 DU or ± 0.7 %).

- Figure 13 will be replaced with the following figure and the figure caption will adjusted accordingly:

[Figure]

- It will be noted in the text that corrected $TCO_{340/305}$ and $TCO_{320/305}$ data agree almost ideally over the time of the eclipse.

- Figure 17 will be replaced with the following figure and the figure caption will adjusted accordingly:

[Figure]

The only change is the color scale of this figure to make it more consistent with the other figures shown above.

**Comment by Referee**

2. Section 4.1 If the spectral response function of the GUV instruments used at Union Glacier were not characterized (lines 250-251), which are the uncertainties related to use the generic response functions on TCO estimations using equation 2?. This issue should be explained in detail.

**Authors' Response**

If generic spectral response functions don't agree with the actual response functions of a GUV, the ratio of GUV measurements and SUV spectra weighted with the generic functions would become dependent on the solar zenith angle (SZA). A good test to determine whether generic response functions are appropriate is therefore to weight the SUV-100 spectra measured during the vicarious calibration of the GUV with these functions and plot the ratio of GUV and weighted SUV data versus SZA. This was done as part of the data analysis and quality control of the calibration that was used for the measurements at Union Glacier. We concluded from this comparison that the SZA-dependence of the ratio is within the normal range of similar comparisons that are routinely executed for GUV instruments for which response functions were measured. Hence, we concluded that the generic spectral response functions for the instrument used at Union Glacier are appropriate and that use of these functions does not increase the uncertainty of ozone data appreciably. This conclusion was not mentioned in the manuscript for the sake of brevity. In response to the referee's concern, we have added this information to the manuscript as shown below (see "Change to manuscript"). To demonstrate that our conclusion is justified, we have plotted the ratios of GUV and SUV measurements established during the calibrations of both GUV radiometers below. We don't think that it is necessary to include these plots also in the paper as be believe that a verbal description is sufficient. We also note that systematic errors in ozone retrievals caused by the uncertainty of the spectral response functions are of minor importance here because the manuscript is about *variations* in TCO, not the *absolute* value. However, the comparison with OMI data (Figure 17 of manuscript) confirms that GUV ozone data a Union Glacier are actually quite accurate.

[Figure]

Ratio of GUV and SUV-100 measurements for data collected during the calibration of the GUV radiometers used at Union Glacier (left) and in Mazatlán and Fort Collins (right), plotted versus the solar zenith angle for the 305 (blue), 313 (green) and 340 (red) nm channels of the GUVs. GUV data are based on one-minute averages that were interpolated to the times when the SUV-100 scanning spectroradiometer measured at the nominal wavelengths of the GUV. Outliers are mostly caused by interpolation uncertainties during scattered-cloud conditions. The ratio of GUV / SUV-100 measurements averaged over 10° SZA intervals is similar for the two instruments and generally within ± 2% of unity. One exception is the ratio for the 305 channel of the GUV used at Union Glacier, which is biased low by about 3 % between SZAs of 70 to 80°. This bias is within the published uncertainty of GUV calibrations, which is 7.5 % (expanded (k=2) uncertainty) for measurements at 305 nm (supplement of Bernhard and Petkov (2019)). Note that the vicarious calibration of the GUV used at Union Glacier was based on 7 days of data while the calibration for the GUV at Mazatlán and Fort Collins was based on 3 days of data. This difference explains the higher point density in the left plot.

**Change to manuscript**

The following text will be added at the end of Section 3.1:

> "To assess whether the use of generic spectral response functions is appropriate, we weighted the SUV-100 Version 2 spectra used for the instrument's calibration in 2015 with these generic functions and compared the weighted irradiances with the contemporaneous measurements of the GUV instrument. If generic and actual functions deviate, the ratio of GUV and SUV-100 measurements would become dependent on the solar zenith angle (SZA) as described in Sect. 4.1. The actual SZA-dependence was similar to that calculated for the GUV radiometer used in Mazatlán and Fort Collins (for which the response functions were measured), suggesting that the use of generic spectral response functions does not appreciably increase the uncertainty of ozone data derived from measurements at Union Glacier."

**Comment by Referee**

3. Sections 5.2.2, 5.3.2 and 5.4.2 In my opinion, these three sections about AOD behaviour during eclipses should be removed since are out of the scope of the manuscript. Additionally, the last paragraph of Section 6 should be accordingly rewritten, and Section 3.3 about Cimel instrument also removed. The authors can work in detail about this topic in the future, providing their results in a new manuscript focused on the measured of AOD during solar eclipses.

**Authors' Response**

While AOD measurements are not essential for understanding ozone variations during a solar eclipse, they are useful because they provide data for the modelling of the spectral irradiance shown in Figures 6 and 14 of the manuscript. AOD data corrected for the solar LD effect also confirm that this correction is accurate. Specifically, LD-corrected data are virtually constant over the time period of the eclipse (Figures 12 and 16) as one would expect. As pointed out in the manuscript, appropriate LD correction is essential for accurate ozone retrievals during the period of the eclipse. Furthermore, unfiltered time series of AOD data, as shown in Figures 6 and 14, are also valuable for indicating periods affected by clouds. The low AODs at Union Glacier also highlight the pristine conditions at this site. While it would be an option to remove parts of the manuscript that discuss AOD and publish these results elsewhere, we feel that it better not to break the observations during the three eclipses apart.

The manuscript also forms the basis of another publication that is currently in preparation and will simulate the spectral irradiance during totality at Mazatlán and Union Glacier with 3-D Monte Carlo model calculations. This planned publication will be similar to that by Ockenfuß et al. (2020) concerning the 2017 total solar eclipse and will also investigate the effects of AOD, the ozone profile, and surface albedo on the radiative transfer during totality. Having all relevant data described and cross-linked in one manuscript is advantageous.

We note that it was already mentioned in the abstract that the manuscript also deals with AOD data: ("In addition to calculating TCO, we also present changes in the spectral irradiance and aerosol optical depth during eclipses and compare radiation levels observed during totality").

I light of these arguments we would prefer keeping the AOD data in the manuscript.

**Change to manuscript**

To emphasize the usefulness of AOD observations, the following will be added to Sect. 3.1:

> "The aerosol optical depth (AOD) was derived from observations of direct spectral irradiance. These data are useful for characterizing atmospheric conditions,

identifying contamination by clouds, validating the LD correction, and providing input parameters for the radiative transfer calculations that complement the measurements."
* * *
**References**

Bernhard, G. and Petkov, B.: Measurements of spectral irradiance during the solar eclipse of 21 August 2017: reassessment of the effect of solar limb darkening and of changes in total ozone, Atmos. Chem. Phys., 19, 4703-4719, https://doi.org/10.5194/acp-19-4703-2019, 2019.

Dahlback, A.: Measurements of biologically effective UV doses, total ozone abundances, and cloud effects with multichannel, moderate bandwidth filter instruments, Appl. Opt., 35, https://doi.org/10.1364/AO.35.006514, 1996.

Ockenfuß, P., Emde, C., Mayer, B., and Bernhard, G.: Accurate 3-D radiative transfer simulation of spectral solar irradiance during the total solar eclipse of 21 August 2017, Atmos. Chem. Phys., 20, 1961-1976, https://doi.org/10.5194/acp-20-1961-2020, 2020.

Piedehierro, A. A., Cancillo, M. L., Serrano, A., Antón, M., and Vilaplana, J. M.: Selection of suitable wavelengths for estimating total ozone column with multifilter UV radiometers, Atmos. Environ., https://doi.org/10.1016/j.atmosenv.2017.04.022, 2017.

Stamnes, K., Slusser, J., and Bowen, M.: Derivation of total ozone abundance and cloud effects from spectral irradiance measurements, Appl. Opt., 30, 4418-4426, https://doi.org/10.1364/AO.30.004418, 1991.